# Reduced nitrogenase efficiency dominates response of the globally important nitrogen fixer *Trichodesmium* to ocean acidification

Ya-Wei Luo [1], Dalin Shi [2], Sven A. Kranz[3], Brian M. Hopkinson[4], Haizheng Hong[2], Rong Shen[2] & Futing Zhang[2]

The response of the prominent marine dinitrogen ($N_2$)-fixing cyanobacteria *Trichodesmium* to ocean acidification (OA) is critical to understanding future oceanic biogeochemical cycles. Recent studies have reported conflicting findings on the effect of OA on growth and $N_2$ fixation of *Trichodesmium*. Here, we quantitatively analyzed experimental data on how *Trichodesmium* reallocated intracellular iron and energy among key cellular processes in response to OA, and integrated the findings to construct an optimality-based cellular model. The model results indicate that *Trichodesmium* growth rate decreases under OA primarily due to reduced nitrogenase efficiency. The downregulation of the carbon dioxide ($CO_2$)-concentrating mechanism under OA has little impact on *Trichodesmium*, and the energy demand of anti-stress responses to OA has a moderate negative effect. We predict that if anthropogenic $CO_2$ emissions continue to rise, OA could reduce global $N_2$ fixation potential of *Trichodesmium* by 27% in this century, with the largest decrease in iron-limiting regions.

[1] State Key Laboratory of Marine Environmental Science and College of Ocean and Earth Sciences, Xiamen University, 361102 Xiamen, Fujian, China. [2] State Key Laboratory of Marine Environmental Science and College of the Environment and Ecology, Xiamen University, 361102 Xiamen, Fujian, China. [3] Department of Earth, Ocean and Atmospheric Science, Florida State University, Tallahassee, FL 32306, USA. [4] Department of Marine Sciences, University of Georgia, Athens, GA 30602, USA. These authors contributed equally: Ya-Wei Luo, Dalin Shi. Correspondence and requests for materials should be addressed to Y.-W.L. (email: ywluo@xmu.edu.cn) or to D.S. (email: dshi@xmu.edu.cn)

Marine N$_2$ fixation conducted primarily by cyanobacteria (diazotrophs) accounts for as much as one half of the input of bioavailable nitrogen (N) to the global ocean[1]. It is thus important to understand how N$_2$ fixation will respond to ocean acidification (OA, i.e., the increase of CO$_2$ concentration and the concomitant decrease of pH in the seawater) caused by the dissolution of anthropogenic CO$_2$ in the ocean[2]. Most previous studies have shown that the growth and N$_2$ fixation of marine diazotrophs, particularly the prominent genus *Trichodesmium*, increased with OA[3]. In contrast, recent studies have reported no significant or even negative effects of OA on diazotrophs[4–7].

The growth enhancement of diazotrophs under OA is often attributed to the downregulation of CO$_2$-concentrating mechanisms (CCM) under high CO$_2$ concentration, which seemingly saves energetic resources for other cellular processes including N$_2$ fixation[8–11]. Hong et al.[7], however, reported that OA inhibited the growth and N$_2$ fixation of *Trichodesmium*, because the beneficial effect of high CO$_2$ concentration was overwhelmed by the negative effect of low pH. Their study suggested that *Trichodesmium* needed to invest additional cellular resources and energy to cope with the stress imposed by low pH (e.g., cytosolic pH disturbance). Regardless of whether OA effects are positive or negative, these studies all highlighted the importance of energy metabolism in the response of *Trichodesmium* to OA, even though they lacked a quantitative understanding of the energy budget.

Iron (Fe) plays a vital role in energy metabolism of *Trichodesmium* and is often a limiting resource for the diazotroph in a large part of the ocean[12–14]. *Trichodesmium* normally allocates a significant portion of intracellular Fe to nitrogenase, an Fe-rich enzyme that catalyzes N$_2$ fixation, and the remaining Fe is used in photosystems and other cellular processes[15–18]. It has been shown that *Trichodesmium* can reallocate intracellular Fe among different cellular processes in response to Fe limitation. For example, under Fe deficiency *Trichodesmium* can compromise on N$_2$ fixation to conserve Fe for photosynthesis[19,20]. Conversely, Fe can be reallocated from photosystems to nitrogenase in Fe-limited *Trichodesmium* to compensate for the decreased nitrogenase efficiency under OA[4,7]. However, it is not fully understood how the reallocation of Fe and energy are quantitatively linked. There still exists a gap between experimental results and model predictions of N$_2$ fixation in the future acidified ocean.

In this study, using experimentally-measured intracellular Fe in photosystems and nitrogenase together with other parameters obtained from the literature, we quantitatively analyze intracellular Fe and energy allocations in *Trichodesmium* in response to OA and examine how they modulate its growth and N$_2$ fixation. The results of these quantitative analyses provide parameterization schemes for an optimality-based *Trichodesmium* cellular model, in which growth rate is maximized by optimizing allocation of intracellular Fe and energy under varying levels of OA and intracellular Fe. By using the model to study different physiological processes, we find that the reduced nitrogenase efficiency dominates the response of *Trichodesmium* to OA. Furthermore, we project that N$_2$ fixation potential by *Trichodesmium* in the global ocean may be reduced by 27% by the end of this century if anthropogenic CO$_2$ emissions continue to rise.

## Results

**Trichodesmium cellular model framework.** We first constructed a framework for the *Trichodesmium* cellular model in which N$_2$ fixation is the only source of N for the diazotroph (Fig. 1). The model uses seawater $p$CO$_2$ and pH, and intracellular Fe as input variables, and allows variable allocation of intracellular Fe and

energy among different cellular processes. It should be noted that although the model is conventionally named as a cellular model, it actually simulates the daily-average response of a filamentous trichome consisting of multiple cells, despite the fact that N$_2$ fixation and photosynthesis in *Trichodesmium* have been shown to be segregated spatially in different cells along a trichome and/or temporally at different time over a diel cycle[21,22].

In the model framework (Fig. 1), the total intracellular Fe quota (Fe discussed hereafter refers to Fe quota, i.e., the cellular Fe to carbon ratio, unless otherwise specified), $Q_{Fe}$, consists of Fe in nitrogenase $\left(Q_{Fe}^{NF}\right)$, photosystems $\left(Q_{Fe}^{PS}\right)$, maintenance $\left(Q_{Fe}^{MT}\right)$ and storage $\left(Q_{Fe}^{ST}\right)$, and the energy produced from the photosystems ($E$) is allocated to CCM ($E^{CCM}$), carbon (C) fixation ($E^{CF}$), N$_2$ fixation ($E^{NF}$), maintenance ($E^{MT}$) and anti-stress against OA ($E^{Ats}$) (Fig. 1). Here, the term maintenance collectively refers to all the incalculable cellular processes that use energy or Fe (such as the tricarboxylic acid cycle and DNA protection[23]). In this model, the amount of Fe allocated to photosystems determines the energy production rate, and the subsequent allocation of energy determines the C fixation rate. The N$_2$ fixation rate also depends on the energy allocated to this process and the amount of Fe allocated to nitrogenase. Therefore, at a given condition, the model potentially has a solution for optimal allocation of intracellular Fe and energy for maximal growth, under which the ratio of C to N fixation equals the elemental stoichiometry of *Trichodesmium* cells and no intracellular Fe and energy are wasted.

To resolve different intracellular Fe pools, $Q^{Fe}$, $Q_{Fe}^{NF}$, and $Q_{Fe}^{PS}$ were determined under different conditions (Table 1) in culture experiments through quantitative measurements of key proteins (see Methods). We then quantitatively estimated $Q_{Fe}^{MT}$ and $Q_{Fe}^{ST}$ as described below.

**Maintenance Fe.** We first estimated $Q_{Fe}^{MT}$ in the acidified low-Fe treatment by assuming no Fe storage $\left(Q_{Fe}^{ST} = 0\right)$ under this highly stressful condition (i.e., $Q_{Fe}^{MT} = Q_{Fe} - Q_{Fe}^{NF} - Q_{Fe}^{PS}$) (Table 1). The maintenance Fe use efficiency (IUE) for growth (IUE$^{MT}$, ratio of carbon-based specific growth rate $g_c$ to $Q_{Fe}^{MT}$) in this treatment was assumed to be constant and applied to estimate $Q_{Fe}^{MT}$ from the growth rates for other treatments (Table 1). It is worth noting that this method could potentially overestimate $Q_{Fe}^{MT}$ because first, $Q_{Fe}^{MT}$ estimated in the acidified low-Fe treatment in fact was its upper bound; second, less Fe may be needed in maintenance under ambient conditions than acidified conditions[7]; and third, comparing high Fe to low Fe conditions, the expression level of Fe-containing proteins involved in maintenance increased relatively less than the increase in growth rate (F. Z., H. H., and D. S., unpublished data). As $Q_{Fe}^{MT}$ was nevertheless small (1.2–9.3% of $Q^{Fe}$, see Table 1), our results should not be affected significantly. In the model simulation, we also used this estimated IUE$^{MT}$ as a constant parameter to calculate $Q_{Fe}^{MT}$ under any condition (Eq. 11 in Methods).

**Fe storage.** With $Q_{Fe}^{MT}$ determined, we then can estimate the Fe storage for all the laboratory treatments: $Q_{Fe}^{ST} = Q_{Fe} - Q_{Fe}^{NF} - Q_{Fe}^{PS} - Q_{Fe}^{MT}$ (Table 1). The Fe storage was quite high in the high-Fe treatments (>80% of $Q_{Fe}$), which was consistent with our observation that $Q_{Fe}$ in *Trichodesmium* grown at even moderate rates was markedly higher than the Fe in nitrogenase and photosystems in those grown at the highest rates (Table 1). In other words, *Trichodesmium* put aside large amount of Fe, even if higher growth rates could be achieved by allocating Fe to metabolism. Such a phenomenon is often the result of luxury Fe uptake and is commonly observed in the field. As Fe is often a

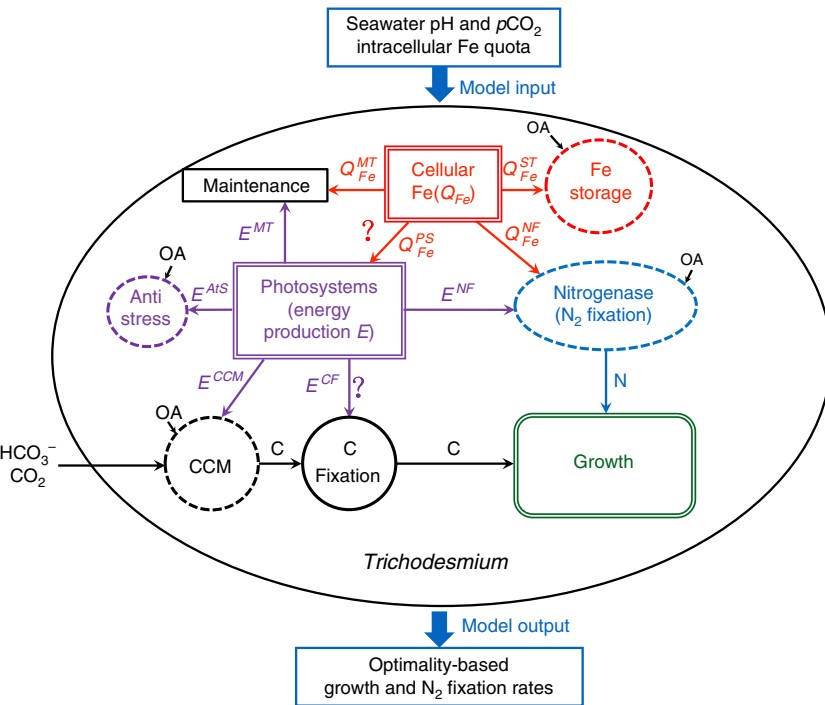

**Fig. 1** *Trichodesmium* cellular model structure. The intracellular Fe (red) and produced energy (purple) are allocated to different cellular processes. The dashed circles and ellipses pointed by OA indicate processes that are influenced by ocean acidification. The two flows with question mark represent unknown parameters to be optimized to maximize *Trichodesmium* growth

---

**Table 1 Treatments and results of *Trichodesmium* manipulation experiments**

| $Fe_T$ (nM) | Fe′ (pM) | Specific growth rate ($d^{-1}$) | $N_2$ fixation [mol N (mol C)$^{-1}$ $d^{-1}$] | POC: PON | Intracellular Fe Quota [μmol Fe (mol C)$^{-1}$] | | | | | | Nitrogenase IUE [mol N (mol Fe)$^{-1}$ $d^{-1}$] |
|---|---|---|---|---|---|---|---|---|---|---|---|
| | | | | | Total ($Q_{Fe}$) | Nitrogen-ase ($Q_{Fe}^{NF}$) | Photo-systems ($Q_{Fe}^{PS}$) | Mainten-ance ($Q_{Fe}^{MT}$) | Metabolism ($Q_{Fe}^*$) | Storage ($Q_{Fe}^{ST}$) | |
| *Ambient (pH = 8.02, pCO₂ ≈ 400 μatm)* | | | | | | | | | | | |
| 10.5 | 32.0 | 0.31 ± 0.01 | 0.056 ± 0.003 | 5.6 | 27.9 | 16.1 ± 1.9 | 6.1 ± 0.6 | (2.6) | 24.8 | 3.1 | 3470 |
| 50.5 | 155 | 0.39 ± 0.01 | | | 88.6 | | | | | | |
| 150 | 461 | 0.49 ± 0.01 | | | 198 | | | | | | |
| 250 | 767 | 0.53 ± 0.02 | | | 287 | | | | | | |
| 300 | 920 | 0.56 ± 0.02 | 0.109 ± 0.002 | 5.1 | 328 | 37.8 ± 4.5 | 13.0 ± 0.2 | (4.8) | 55.5 | 272.8 | 2880 |
| *Acidified (pH = 7.82, pCO₂ ≈ 700 μatm)* | | | | | | | | | | | |
| 32.5 | 39.5 | 0.22 ± 0.02 | 0.032 ± 0.001 | 5.9 | 32.5 | 27.0 ± 5.1 | 3.7 ± 0.6 | 1.86 | 32.5 | 0 | 1200 |
| 125 | 153 | 0.30 ± 0.01 | | | 87.8 | | | | | | |
| 380 | 463 | 0.43 ± 0.04 | | | 198 | | | | | | |
| 765 | 931 | 0.46 ± 0.02 | 0.095 ± 0.014 | 5.1 | 331 | 43.6 ± 3.4 | 16.7 ± 0.8 | (3.9) | 64.3 | 266.7 | 2180 |

Errors denote 1 s.d. (n = 3)
$Fe_T$ concentration of total dissolved Fe in medium, Fe′ concentration of dissolved inorganic Fe in medium

---

limiting resource in the oceans and its supply is episodic, marine phytoplankton including *Trichodesmium* take up more Fe than their metabolic requirements and store this excess Fe for later use even under intermediate Fe limitation. For example, two studies[15,24] have both reported linear increases in growth rate of *Trichodesmium* over a range of low $Q_{Fe}$ values, consistent with a lack of Fe storage under severe limitation. However, marginal increases in $Q_{Fe}$ with further increases in inorganic Fe (Fe′) (intermediate Fe limitation) result in lesser increases in growth rate. We interpret these earlier data to suggest that only a small portion of the marginal increase in $Q_{Fe}$ is used for growth and that perhaps 80% of this marginal increase is used for storage; this interpretation is consistent with our model results. Although such high $Q_{Fe}^{ST}$ is feasible in *Trichodesmium* given the high Fe storing capacity of Dps[23] and ferritin[25] (260 and 4500 Fe atoms per protein molecule, respectively) (Supplementary Tables 1 and 2,

Supplementary Note 1), it remains unclear why this diazotroph stores such a high amount of Fe particularly under Fe-replete conditions, which warrants further investigation.

We then extrapolated the results from our culture experiments to construct a model scheme for Fe storage quota. Fe storage is expressed as a constant portion ($f^{ST}$) of excess cellular Fe (the $Q_{Fe}$ above a threshold $Q_{Fe}^c$), while the residual is defined as metabolic Fe ($Q_{Fe}^*$, sum of $Q_{Fe}^{NF}$, $Q_{Fe}^{PS}$, and $Q_{Fe}^{MT}$) (Eq. 3 in Methods). We found that $f^{ST} = 90\%$, while the threshold $Q_{Fe}^c$ increased under acidified conditions (Fig. 2a). This increase in the threshold $Q_{Fe}^c$ is most likely caused by higher Fe requirements for metabolic processes under acidified conditions. Thus, we parameterized $Q_{Fe}^c$ as a function of pH by further assuming that the relative changes of $Q_{Fe}^c$ and seawater [$H^+$] were proportional (Eqs. 2 and 3 in Methods). Although the pH of seawater medium was used in our calculations, the cellular machinery was located in the cytosol or

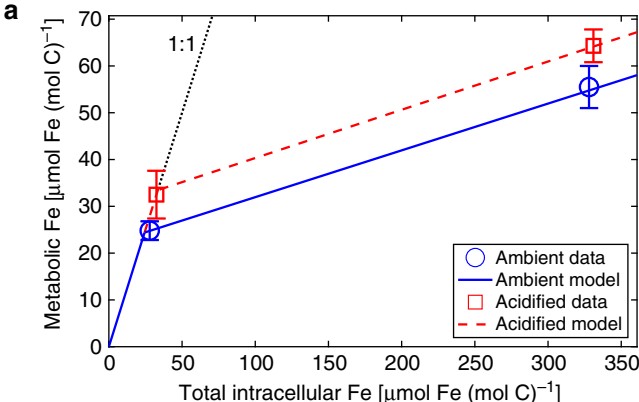

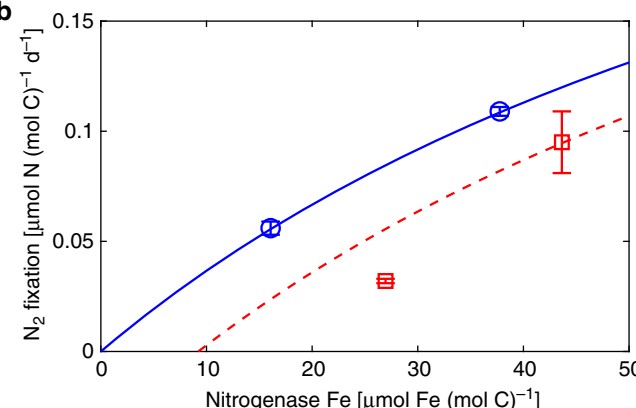

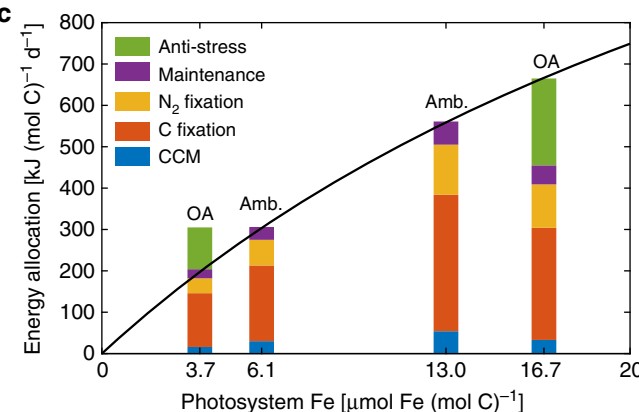

**Fig. 2** Illumination of model schemes. **a** Fe storage model scheme and **b** nitrogenase Fe-determined N$_2$ fixation rate model scheme compared with the experimental data under ambient (blue) and acidified (red) conditions. **c** Energy production model scheme (black line) and the estimates (colored bars) of energy allocation to different cellular processes in the culture experiments under ambient (Amb.) and acidified (OA) conditions. For experimental data, error bars represent the s.d. of biological replicates ($n = 3$)

the thylakoid membranes of the organism where the pH was not necessarily the same as in the medium. The extent of pH decrease and accordingly the relative increase of [H$^+$] in cytosol were, however, close to those in seawater[7].

**Nitrogenase efficiency and N$_2$ fixation.** Nitrogenase IUE (the ratio of N$_2$ fixation rate to $Q_{Fe}^{NF}$) under ambient conditions fell in a range that was consistent with the previous estimates for *Trichodesmium*[26], and decreased with increasing $Q_{Fe}^{NF}$ (Table 1),

which may reflect a relationship between N$_2$ fixation rate and gradually saturating nitrogenase concentration. To empirically reproduce this phenomenon, we adopted the Monod equation to parameterize N$_2$ fixation rate as a function of $Q_{Fe}^{NF}$ under ambient conditions (Fig. 2a and Eq. 13 in Methods).

The observed nitrogenase IUE was reduced under acidified conditions (Table 1), which may be due to a higher ratio of electron allocation to H$^+$, instead of N$_2$, for H$_2$ evolution[4,27]. The nitrogenase IUE declined more in the low-Fe treatment (65%) than in the high-Fe treatment (24%) (Table 1) (discussed later). We hence set up a parameterization scheme to reflect our observations that nitrogenase IUE was inversely proportional to seawater [H$^+$], with the reduction inversely proportional to $Q_{Fe}^{NF}$ (Fig. 2b and Eq. 14 in Methods).

To match the observed N$_2$ fixation rate of the low-Fe acidified treatment, only a $Q_{Fe}^{NF}$ of ~19 μmol Fe (mol C)$^{-1}$ is required, which is much lower than the observed value of 27.0 μmol Fe (mol C)$^{-1}$ (Fig. 2b). However, the samples for intracellular Fe analyses were collected at mid-day when photosynthesis is downregulated and nitrogenase activity was high[21]. *Trichodesmium* may significantly reduce photosynthesis to reallocate Fe to nitrogenase at mid-day[4], especially in the highly-stressful acidified low-Fe treatment, in compensation for the loss of N$_2$ fixation efficiency at low pH. The extent of the increase in nitrogenase averaged over a diel cycle may not be as large as those appeared in the mid-day samples. This has been observed in a previous study, in which OA caused *Trichodesmium* grown under low Fe conditions (40 pM Fe') to increase NifH (the nitrogenase reductase of the nitrogenase complex) by 47% at mid-day but by 37% on average on a daily basis[4].

**CCM energy consumption.** To estimate energy consumption rate for the *Trichodesmium* CCM, we considered a simple scheme including that first, a portion ($f^{BC}$) of total inorganic carbon uptake (Ci) into the cell is from energy-consuming bicarbonate (HCO$_3^-$) transport; second, passive CO$_2$ diffusion contributes the rest of Ci; and third, a portion ($l_k$) of Ci leaks out of the cell as CO$_2$ (Supplementary Figure 1). The CO$_2$ passively diffusing into cytoplasm is converted to HCO$_3^-$ at the nicotinamide adenine dinucleotide phosphate (NADPH) dehydrogenase complex located at the thylakoid membrane[28]. The mechanistic details of this system have not been determined, but the overall energy cost is thought to be small. NADPH consumption supporting proton removal is likely coupled to ATP generation resulting in a low net energy cost[29]. We thus neglect this CO$_2$ uptake dependent energy consumption and only consider the energy consumption associated with HCO$_3^-$ transport. Transport of 1 mol HCO$_3^-$ costs 1 mol ATP[30] (approximately 50 kJ[31]), and building HCO$_3^-$ transporters and carboxysomes also requires energy, which, however, is hard to quantify and is assumed to cost additional 20% energy. Altogether the cost is estimated at 60 kJ per mol HCO$_3^-$ transported. Using $f^{BC} = 80\%$ and $l_k = 50\%$ in *Trichodesmium*[9,32,33], 1 mol C fixation requires 2.0 mol Ci, in which 1.6 mol is from HCO$_3^-$ transport, and accordingly 96 kJ of energy (Table 2). It is worth noting that even if building HCO$_3^-$ transporters and carboxysomes costs four times more energy (i.e., 100%), the CCM energy cost rate would increase from 96 to 160 kJ (mol C)$^{-1}$, which is still small compared to the total energy cost (Table 2).

As seawater acidifies, the potential for diffusive CO$_2$ flux increases approximately proportionally with dissolved [CO$_2$] in seawater[34], and accordingly the cellular demand for transport of HCO$_3^-$ decreases. In addition, the increase of dissolved [CO$_2$] in seawater can reduce the cross-membrane [CO$_2$] gradient and hence CO$_2$ leakage. However, the reduction of CO$_2$ leakage under

**Table 2 Energy consumption rates of *Trichodesmium* cellular processes under ambient condition**

|  | ATP hydrolysis | NADPH oxidation | Energy consumption rate[a] | References |
|---|---|---|---|---|
| C fixation | 3 per C | 2 per C | 590 kJ (mol C)$^{-1}$ | Raven et al.[60] |
| CCM | 1.92 per C[b] |  | 96 kJ (mol C)$^{-1}$ |  |
| N fixation include: | 9 per N | 3 per N | 1,110 kJ (mol N)$^{-1}$<br>= 198 kJ (mol C)$^{-1\,c}$ |  |
| N$_2$ assimilation to NH$_4^+$ | 8 per N | 2 per N |  | Flores and Herrero[61] |
| NH$_4^+$ assimilation to glutamate | 1 per N | 1 per N |  | Flores et al.[62] |
| Maintenance and other processes |  |  | 100 kJ (mol C)$^{-1}$ | ~10% of total energy consumption |
| Total |  |  | 984 kJ (mol C)$^{-1}$ |  |

[a]Energy consumption rate was estimated based on the free energy of 50 kJ (mol ATP hydrolysis)$^{-1}$ and of 220 kJ (mol NADPH oxidation)$^{-1\,31}$
[b]Assuming 80% HCO$_3^-$ use and 50% CO$_2$ leakage[9], and a transport cost of 1.2 ATP per HCO$_3^-$ [30]
[c]Using C:N = 5.6 to convert energy consumption of N fixation to C unit

OA is fairly small[9,33], likely because, as in other cyanobacteria, CO$_2$ concentration within *Trichodesmium* cells is among the highest in phytoplankton[35,36], and thus the increase of extra-cellular [CO$_2$] should not change the cross-membrane [CO$_2$] gradient substantially. We thus parameterized a scheme for the CCM energy consumption (Eq. 6 in Methods), in which a doubling of $p$CO$_2$ reduced CO$_2$ leakage by 10%, and, together with the reduced HCO$_3^-$ transport, decreased the CCM energy consumption by 32%. This estimate could depend on the choice of initial $f^{BC}$ before changing $p$CO$_2$. For instance, the initial $f^{BC}$ could be smaller under a lower growth rate as a larger fraction of Ci can be met by CO$_2$ diffusion. As shown by a model sensitivity test (Supplementary Figure 2), for example, for a doubling of $p$CO$_2$ the CCM energy consumption can be reduced ~50% if the initial $f^{BC}$ = 70%, and can be reduced to zero if the initial $f^{BC}$ < 50%. Nevertheless, as discussed below, the energy consumed by CCM is small and its saving does not impact model results substantially.

**Other energy costs and production under ambient conditions.** Energy consumption rates for C fixation and N$_2$ fixation were estimated based on theoretical energy requirements for ATP hydrolysis and NADPH oxidation of these reactions, and maintenance was assumed to cost ~10% of total energy (Table 2). From the measured rates of *Trichodesmium* growth and N$_2$ fixation, we estimated total cellular energy consumption, which equals energy production assuming no energy waste, for the ambient low- and high-Fe treatments (Fig. 2c). Similar to nitrogenase, the IUE of photosystems for energy production decreases with the increase of $Q_{Fe}^{PS}$, and we adopted a Monod-like equation to represent the relationship between energy production rate and $Q_{Fe}^{PS}$ (Fig. 2c and Eq. 4 in Methods).

**Anti-stress energy consumption under acidified conditions.** If OA does not change the IUE of photosystems, using the parameterization established above (Eq. 4 in Methods), the energy produced in the acidified high-Fe treatment would be 46% more than the total requirement for the CCM, C and N fixations, and maintenance (Fig. 2c). We considered this excess energy as the cost for anti-stress against OA, which, for example, is needed for maintaining cytosolic pH homeostasis[7]. We subsequently scaled the observations and parameterized the anti-stress energy proportional to C-based specific growth rate and relative change of seawater [H$^+$] (Eq. 7 in Methods).

To simulate the energy production rate in the acidified low-Fe treatment, 6.1 μmol Fe (mol C)$^{-1}$ is needed by the photosystems (Fig. 2c). This was significantly higher than the measured 3.7 μmol Fe (mol C)$^{-1}$ (Table 1) (at mid-day), therefore also

suggesting that the difference between daily average Fe in nitrogenase and photosystems can be less than it appeared in the mid-day samples, as discussed above.

**Cellular model simulation.** The parameterization schemes established above were integrated into the cellular model framework (Fig. 1). As shown by the model equation sequence in Methods, given the input variables of $Q_{Fe}$ and seawater $p$CO$_2$ and pH, *Trichodesmium* growth and N$_2$ fixation rates in the model are a function of only two parameters, i.e., $f_{Fe}^{PS}$, the fraction of metabolic Fe $\left(Q_{Fe}^*\right)$ allocated to photosystems, and $f_E^{CF}$, the fraction of produced energy allocated to C fixation, which are solvable by maximizing the growth rate (see Methods).

The simulated growth and N$_2$ fixation rates increase with $Q_{Fe}$. Above the Fe threshold (at which the cells start to store Fe) the rate of increase is reduced (Fig. 3a, b). The model accurately reproduces most of the measured growth rates under both the ambient and acidified conditions (Fig. 3a). The model also reproduces the N$_2$ fixation rates well, other than a slight overestimation for the acidified low-Fe treatment (Fig. 3b). This likely is because the model uses a constant C:N ratio for biomass, yet the experimentally measured ratio varied (Table 1). The modeled Fe allocation to nitrogenase and photosystems generally reproduces the measurements accurately, except for the acidified low-Fe treatment where the modeled Fe in photosystems is substantially higher than the measurement values (Fig. 3c, d). This is likely due to the stoichiometric differences in the respective acclimations discussed above.

The simulation results over a range of seawater pH (and concomitant change in $p$CO$_2$, See Methods) and $Q_{Fe}$ show that OA has a large impact on *Trichodesmium* when $Q_{Fe}$ is low (Figs 3e, f). When $Q_{Fe}$ > 50 μmol Fe (mol C)$^{-1}$, OA changes *Trichodesmium* growth rate by no more than ±20% with a pH change of ±0.2 unit (Fig. 3f).

It should be noted that although linear or Monod-like equations are assumed and applied in the cellular model, N$_2$ fixation rates and intracellular Fe allocations which the model is parameterized and tested against are derived experimentally at only two Fe levels (Fig. 2). As the actual underlying relationships are unknown, it is possible that our model results may be off to a certain degree in particular at moderate levels of Fe. Nevertheless, the model reasonably fitted the measured growth rates observed at multiple Fe concentrations (i.e., 5 under ambient conditions and 4 under acidified conditions) (Fig. 3a), indicating that the assumed relationships in our model are reasonably sound. In addition, the N$_2$ fixation rates that the model fitted to were experimentally measured with the acetylene reduction method[37], using a fixed ratio of 4:1 to convert ethylene production to N$_2$ fixation. The ratio however can vary between 3:1 and 4:1[38], and in

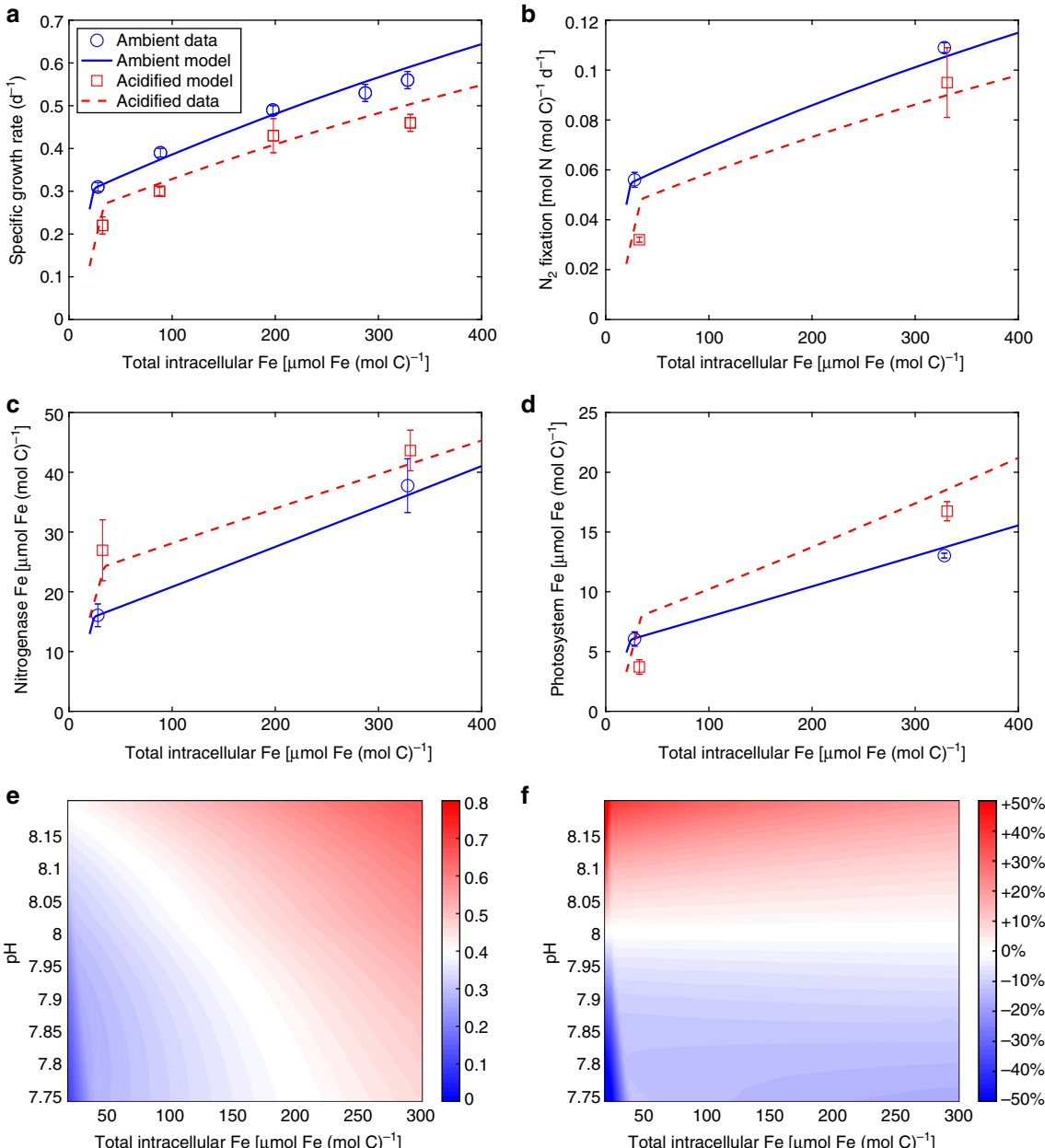

**Fig. 3** Cellular model simulation results. **a** Specific growth rate, **b** $N_2$ fixation rate, **c** amount of Fe allocated to nitrogenase, and **d** amount of Fe allocated to photosystems under ambient (blue), and acidified (red) conditions, compared to the observations. **e** Growth rate and **f** the relative change of growth rate compared to that under pH 8.02 over ranges of seawater pH and intracellular Fe levels. For experimental data, error bars represent the s.d. of biological replicates ($n = 3$)

fact can be lower in *Trichodesmium* grown under ambient conditions than under acidified conditions[4,7]. Therefore, the actual $N_2$ fixation rates under acidified conditions may be overestimated relative to those under ambient conditions, which would result in an underestimate of the OA impact on *Trichodesmium* $N_2$ fixation by up to 33%.

**Comparison of different OA effects on *Trichodesmium*.** Model experiments with a decrease of 0.2 pH unit and a concomitant increase in $p\mathrm{CO_2}$ (See Methods) show that within a $Q_{Fe}$ range of 20–300 μmol Fe (mol C)$^{-1}$, CCM downregulation alone increases *Trichodesmium* growth rate by only ~0.6%, while anti-stress energy consumption and reduced nitrogenase efficiency decrease

growth by ~11% and 18–46%, respectively (Fig. 4a). The impact of the reduced nitrogenase efficiency diminishes gradually with increasing $Q_{Fe}$, because, as discussed above, the reduction of nitrogenase efficiency becomes proportionally smaller.

These model experiments reveal that the negative effect of reduced nitrogenase efficiency is more significant than that of the anti-stress energy consumption, because most of the intracellular metabolic Fe of *Trichodesmium* is in nitrogenase and much less Fe is in the photosystems (Table 1 and other studies[15,16]). The simulated optimal reallocations of the intracellular resources demonstrate that when the anti-stress energy consumption is the only OA effect, a relatively small portion of nitrogenase Fe (~15%) is reallocated to the photosystems, consequently increasing photosystem Fe greatly by ~40% (Fig. 4b). As a result, the

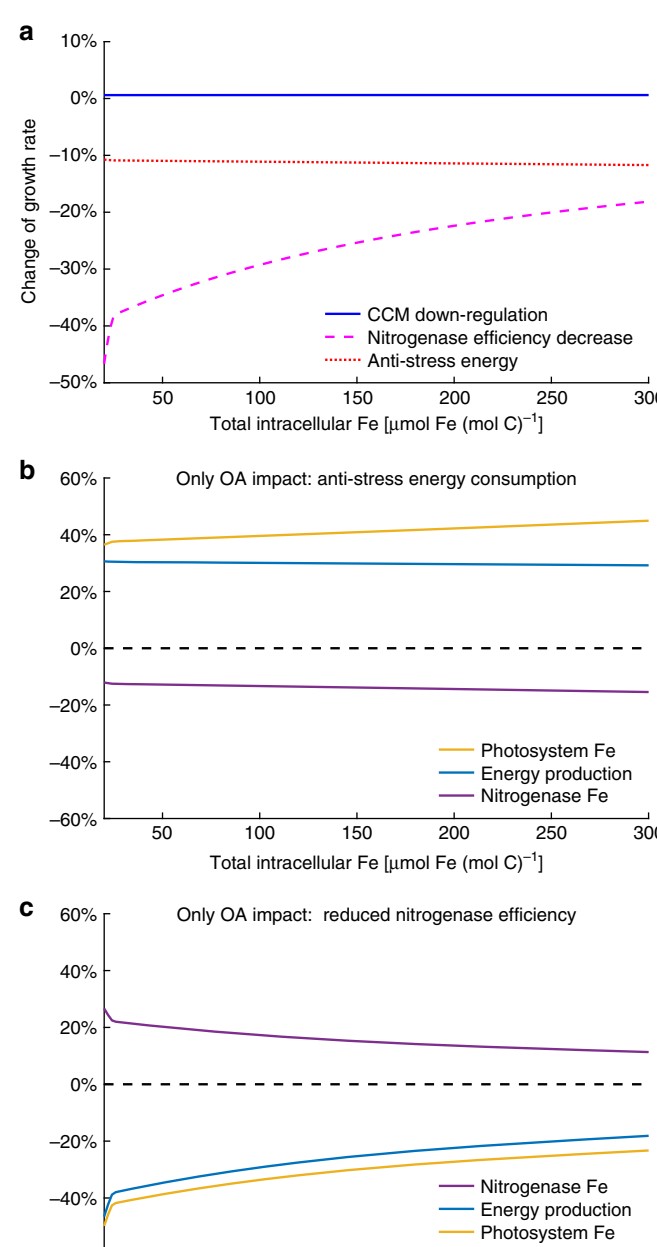

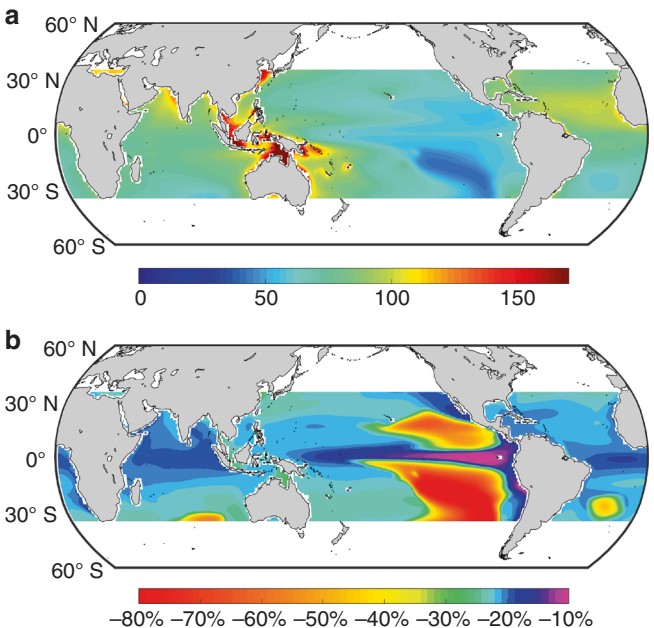

**Fig. 5** The projected *Trichodesmium* N$_2$ fixation potential. **a** Results [mmol N (mol C)$^{-1}$ d$^{-1}$] in 1990s and **b** relative change from 1990s to 2090s

**Projection of global N$_2$ fixation potential by *Trichodesmium*.** The logarithms of in-situ *Trichodesmium* Fe quota and the modeled surface Fe' (from the Community Earth System Model-Biogeochemistry, CESM-BGC) shows a strong linear relationship ($R^2 = 0.65$), with the regression line [Log($Q_{Fe}$) = −0.17 + 0.83Log(Fe')] close to that obtained from a laboratory experiment[4] (Supplementary Figure 3a). The projected *Trichodesmium* $Q_{Fe}$ in the global ocean using this relationship generally matches the experimental observations (Supplementary Figure 3b).

Taking into consideration Fe limitation and $p$CO$_2$/pH only, we estimate *Trichodesmium* N$_2$ fixation potential in the 1990s at 76 ± 20 (mean ± s.d.) mmol N (mol C)$^{-1}$ d$^{-1}$, with the highest potential in the oceanic regions near the Indonesian archipelago and north of Australia and the lowest potential in the southeastern and northeastern subtropical Pacific (Fig. 5a). We also project a decrease of *Trichodesmium* N$_2$ fixation potential from 1990s to 2090s by 27 ± 15% (mean ± s.d.) under the Representative Concentration Pathway (RCP) 8.5, a scenario in which anthropogenic greenhouse gas (including CO$_2$) emissions continue to rise throughout the 21st century (Fig. 5b). The regions with the largest decrease overlap with those where Fe is limiting and the N$_2$ fixation potential is low, i.e., the southeastern and northeastern subtropical Pacific. In most of the other oceanic regions, the N$_2$ fixation potential decreases by ~20%.

## Discussion

In the present study, we systematically and quantitatively evaluated the impact of OA on the prominent marine diazotroph *Trichodesmium*, by analyzing experimental data and establishing a cellular model that takes into account OA effects on a suite of cellular processes: CCM downregulation, anti-stress energy consumption, reduced nitrogenase efficiency, and increased threshold for Fe luxury uptake. The model was well constrained by the experimental observations, particularly $Q_{Fe}$ and the nitrogenase and photosystem proteins, and by theoretical estimates of energy consumptions of cellular processes. With these constraints, under given conditions of OA and $Q_{Fe}$, the model only depended on two parameters (the fraction of energy allocated to C fixation, and the fraction of Fe allocated to photosynthesis), which can be

**Fig. 4** Comparison of modeled individual impact of ocean acidification (OA) on *Trichodesmium*. The pH changes from 8.02 to 7.82 and $p$CO$_2$ changes accordingly in the simulations. **a** Relative changes of growth rate when the model enables each of the OA impacts. Also shown the relative changes of nitrogenase Fe, photosystems Fe, and energy production when the model enables **b** anti-stress energy or **c** the reduced nitrogenase efficiency as the only OA impact

energy production increases by >30% (Fig. 4b), which largely compensates for the anti-stress energy requirements, and therefore the decrease of the growth rate is insignificant (Fig. 4a). However, when the reduced nitrogenase efficiency is the only OA effect, reallocation of substantial Fe from the photosystems (23–50%) results in a small increase in nitrogenase (11–27%) (Fig. 4c). This allocation barely compensates for the reduced nitrogenase efficiency, and meanwhile considerably reduces energy production (18–46%) (Fig. 4c), consequently leading to a greatly reduced growth rate (Fig. 4a).

determined by optimally allocating intracellular Fe and energy to maximize the growth rate of *Trichodesmium*. The results demonstrated that *Trichodesmium* cannot cope with the cumulative stresses imposed by OA and hence may decline in a future, more acidic ocean.

The modeled optimal allocations of intracellular Fe and energy, and therefore the modeled maximal growth and $N_2$ fixation rates, relied mainly on the values of those parameters explicitly or implicitly related to energy consumption, IUE, and the strength of the OA impacts. The major consumption of cellular energy by C and $N_2$ fixation was constrained by free energy changes of those biochemical reactions (Table 2). We estimated that the CCM required only ~10% of total cellular energy (Table 2 and Fig. 2c), which is approximate as inorganic carbon transport processes are not fully understood. For example, CCM costs could be even lower if the $HCO_3^-$ transport were driven by a gradient of $Na^+$ or $H^+$[39]. We also estimated that a doubling of $pCO_2$ can reduce CCM energy consumption by ~30% and therefore change total cellular energy consumption by only ~3%. Additionally, under Fe-limitation, the realized increase in growth rate would be even less than 3%, because the increase in $N_2$ fixation due to the energetic benefit of CCM downregulation would require reallocation of Fe from photosystems to nitrogenase, which in turn would lead to a lower energy production, partially offsetting the benefit of CCM downregulation. Finally, as discussed above, the increase in environmental $CO_2$ may not substantially reduce cellular $CO_2$ leakage in *Trichodesmium*. Taking into consideration all these factors, our model projections reveal that the benefit of a CCM downregulation for enhanced growth of *Trichodesmium* is very limited (0.6% for a doubling of $pCO_2$) (Fig. 4a), which is comparable with an experimental observation that at a given pH a doubling of $pCO_2$ from 400 to 800 μatm led to an increase of *Trichodesmium* growth rate by only 0.8–3.8%[7]. In contrast, an OA stimulated increase in *Trichodesmium* growth and $N_2$ fixation rates of 10–94% and 35–317%, respectively, as previously reported[8–11,40,41], would require the CCM energy consumption rate ~4–40 fold higher than the level estimated in this study, making the CCM cost 36–81% of total cellular energy. Such a considerable energetic expenditure on the CCM would seem infeasible, given the large, competing energetic demands of other processes (e.g., C and $N_2$ fixation) in *Trichodesmium*, even not to mention the energy requirement for coping with the negative impacts of OA. Thus, the positive effect of OA reported previously is likely caused by some of the artificial growth conditions applied that supported overall suboptimal growth rates and potentially altered key physiological processes in *Trichodesmium*[4,7].

The anti-stress energy requirement estimated based on the culture experiments was scaled in the model to other conditions, assuming that it was proportional to growth rate and seawater $[H^+]$, which however remains to be verified. Nevertheless, as discussed above, the model was not very sensitive to the rate of the anti-stress energy consumption as energy can be generated to meet this need by reallocating a small amount of cellular Fe to the photosystems. This, however, assumed that light was not limiting. Our model experiment showed that when cellular energy production is light-limited (see Methods), the anti-stress energy requirement would further reduce *Trichodesmium* growth rate at high intracellular Fe (Supplementary Figure 4). In our culture experiments, however, *Trichodesmium* did not appear to be limited by a light intensity of ~80 μmol photons $m^{-2} s^{-1}$, which was relatively low compared to that in the surface water of the tropical and subtropical open oceans where *Trichodesmium* inhabits. Future experiments with *Trichodesmium* grown under different light and Fe levels are necessary for a better

understanding of the impact of OA on the energy limitations of the diazotroph.

Our model revealed that the response of *Trichodesmium* to OA was dominated by the reduced nitrogenase efficiency. The model results, particularly the more significant adverse effect of OA on $N_2$ fixation under Fe limitation, are largely based on an inverse relationship between $Q_{Fe}$ and the OA-induced reduction of nitrogenase efficiency observed in our culture experiments (Table 1). Although the underlying mechanisms remain unclear, it is possible that under low-Fe conditions OA caused an overall higher degree of stress to cells and thus resulted in a lower level of cellular ATP. Reduced energy supply, in addition to decreased cytosolic pH, may further reduce nitrogenase efficiency by increasing the production of $H_2$ at the expense of $NH_3$, as previously observed in *Azotobacter vinelandi*[27,42]. In addition, at the contemporary rate of increasing atmospheric $pCO_2$ and accordingly seawater acidification, it is possible that *Trichodesmium* may evolutionarily adapt to better cope with the decrease in nitrogenase efficiency. The significant luxury uptake and subsequent storage of Fe by *Trichodesmium* (90% of cellular Fe above a threshold in this study), when the Fe supply is high, likely reflect its response to natural fluctuations in Fe availability by storing Fe for later use. On a longer time-scale, it is possible that the diazotroph may evolve to optimally allocate its cellular Fe, investing more in metabolism than in storage to compensate for the reduced nitrogenase efficiency. This phenomenon has already been found in our study where the Fe storage started at a higher threshold of intracellular Fe (i.e., more Fe used in metabolic processes) under OA (Fig. 2a), although the full extent to which this can compensate for the negative effects of OA is unclear. If the compensation becomes increasingly significant in the future, OA effect on *Trichodesmium* $N_2$ fixation would be less pronounced than that projected in this study, particularly in areas with high Fe supply. Therefore, the mechanism and the degree of the impact of OA on nitrogenase efficiency and Fe storage, possibly in the context of long-term adaptation, seems to be key questions in the future for better predicting the change of marine $N_2$ fixation.

The incorporation of the cellular model we developed into biogeochemical models would help to improve our predictions for $N_2$ fixation and other biogeochemical cycles in the future acidified ocean. Our model used $Q_{Fe}$ as one of the input variables so that it could be constrained more directly by our experimental data and meanwhile avoided simulating the rather complex and not fully understood Fe chemistry in seawater[43]. A strong correlation between the measured *Trichodesmium* $Q_{Fe}$ and the Fe' predicted by an earth system model (Supplementary Figure 3) allowed the extrapolation from our cellular model to the global ocean (Fig. 5b). It is thus promising that our cellular model can be easily connected with large-scale models.

Caution should be taken when applying the model to diazotrophic species/strains other than the one studied here. It should also be noted that the model only represents intracellular Fe and energy allocations, and therefore implicitly assumes that other resources, such as phosphorus[12], temperature[44], cobalt, and vitamins[45], do not limit *Trichodesmium* growth. Growth of *Trichodesmium* is often limited by the deficiency of phosphorus in surface seawater[12], which is expected to intensify as a result of augmented water column stratification caused by global warming. Additionally, a recent study shows that the optimal thermal range of *Trichodesmium* becomes wider under higher $pCO_2$ and thus increasing temperature can help the diazotroph to alleviate the impact of OA[46]. Therefore, changes in these factors in the future ocean may modulate the OA effect and hence could change our model projections. More manipulative OA experiments on

**Table 3 Model variables and parameters**

| Symbol | Unit | Description | Value |
|---|---|---|---|
| *Input variables* | | | |
| $Q_{Fe}$ | μmol Fe (mol C)$^{-1}$ | Total intracellular Fe quota | |
| pH | | pH value in medium | |
| $pCO_2$ | μatm | Partial pressure of medium dissolved $CO_2$ | |
| *Parameters to be optimized* | | | |
| $f_{Fe}^{PS}$ | | Fraction of metabolic Fe allocated to photosystems | |
| $f_E^{CF}$ | | Fraction of produced energy allocated to C fixation | |
| *Constant parameters* | | | |
| $pH_{bsl}$ | | Medium pH value under baseline condition | 8.02 |
| $pCO_{2,bsl}$ | μatm | Medium $pCO_2$ under baseline condition | 400 |
| $Q_{Fe,bsl}^c$ | μmol Fe (mol C)$^{-1}$ | Critical $Q_{Fe}$ for Fe luxury uptake under baseline condition | 24.4 |
| $OA^{ST}$ | | Coefficient representing the strength of OA impact on Fe storage threshold $Q_{Fe}^c$ | 0.71 |
| $f^{ST}$ | | Portion of luxury Fe uptake | 90% |
| $E_{max}$ | kJ (mol C)$^{-1}$ d$^{-1}$ | Maximum cellular energy production rate | 2060 |
| $K_{Fe}^{PS}$ | μmol Fe (mol C)$^{-1}$ | Half-saturating coefficient for energy production | 35 |
| $ec^{CF}$ | kJ (mol C)$^{-1}$ | Energy consumption rate of carbon fixation | 590 |
| $f^{BC}$ | | Contribution of $HCO_3^-$ to total inorganic carbon uptake under baseline condition | 80% |
| $l_k$ | | Percentage of total inorganic carbon uptake leaked as $CO_2$ | 50% |
| $OA^{lk}$ | | Coefficient representing relative reduction of $CO_2$ leakage with increasing medium $CO_2$ | 0.1 |
| $ec^{CCM}$ | kJ (mol C)$^{-1}$ | Energy consumption rate of $HCO_3^-$ transportation | 60 |
| $ec^{AtS}$ | kJ (mol C)$^{-1}$ | Coefficient representing energy consumption rate for anti-stress | 780 |
| $ec^{MT}$ | kJ (mol C)$^{-1}$ | Energy consumption rate of maintenance | 90 |
| $ec^{NF}$ | kJ (mol N)$^{-1}$ | Energy consumption rate of $N_2$ fixation | 1 110 |
| $IUE^{MT}$ | mol C (μmol Fe)$^{-1}$ d$^{-1}$ | Fe use efficiency in maintenance | 0.12 |
| $Q_{Fe,ref}^{NF}$ | μmol Fe (mol C)$^{-1}$ | Reference nitrogenase Fe used in representing impact of pH on nitrogenase efficiency | 25 |
| $NF_{max}$ | mol N (mol C)$^{-1}$ d$^{-1}$ | Maximum $N_2$ fixation rate | 0.37 |
| $K_{Fe}^{NF}$ | μmol Fe (mol C)$^{-1}$ | Half-saturating coefficient for $N_2$ fixation | 91 |
| $r_N^C$ | mol C (mol N)$^{-1}$ | Carbon to nitrogen ratio of *Trichodesmium* cell | 5.6 |

*Trichodesmium* with these limiting factors are thus needed in order to frame a full picture of the mechanisms and improve accuracy of the projections.

## Methods

**Laboratory experimental data**. Growth and $N_2$ fixation rates of *Trichodesmium* at varying Fe concentrations (Fe$_T$) under ambient (pH = 8.02, $pCO_2 \approx 400$ μatm) and acidified conditions (pH = 7.82, $pCO_2 \approx 700$ μatm) are obtained from Hong et al.[7], in which *Trichodesmium* were pre-acclimated to experimental conditions for 2–3 months before the rates were measured. Inorganic Fe concentration (Fe′) (pM) was estimated from Fe$_T$, pH, light, temperature, and concentration of the chelating agent EDTA in the medium[47]. The intracellular Fe quota ($Q_{Fe}$) [μmol Fe (mol C)$^{-1}$] was calculated based on a regression between the logarithms of Fe′ and $Q_{Fe}$ [i.e., Log($Q_{Fe}$) = 0.734 Log (Fe′) + 0.341][4], which is observed previously with the same organism cultured under the same experimental conditions as in the present study.

Following Shi et al.[4], cells from each one of the triplicate cultures grown under different conditions (i.e., ~35 or ~925 pM Fe′ under ambient or acidified conditions) were collected at midday of the photoperiod for quantifying expression of nitrogenase and photosynthetic Fe-containing proteins including: NifH, PsbA, the D1 protein of photosystem II (PSII), PetC, a key subunit of cytochrome (cyt) b6/f complex, and PsaC, the core subunit of photosystem I (PSI), in unit of μmol per g total cellular protein, by immunoblot analyses. Briefly, steady-stately growing cells were collected by filtration onto 3 μm polycarbonate membrane filters (Millipore), flash frozen in liquid nitrogen, and then stored at −80 °C for later analysis. Proteins were extracted and denatured in an extraction buffer (2% SDS, 10% glycerol, and 50 mMTris at pH 6.8; 1% β-mercaptoethanol was added after protein quantification) with heating at 100 °C for 10 min. Insoluble material was pelleted by centrifugation at 12,000 g for 10 min, and total protein in the supernatant was quantified using the bicinchoninic acid (BCA) assay (Pierce, Thermo Scientific). Equivalent amounts of total protein (10 μg for NifH and PetC; 5 μg for PsaC; 1 μg for PsbA) was separated on a 12% SDS polyacrylamide gel for 20 min at 80 V followed by 60 min at 120 V in 1 × SDS running buffer, and then transferred onto a PVDF membrane in ice-cold transfer buffer (25 mM Tris, 192 mM glycine and 2.5% methanol) for 20 min at 300 mA. The membrane was then blocked for 1 h in TBST buffer (Tris-buffered saline with 0.25% Tween 20) containing 5% nonfat milk, followed by 1–2 h incubation with primary antibody (Agrisera Antibodies, Sweden: NifH, Art no. AS01 021 A; PsbA, Art no. AS05 084; Pet C, Art no. AS08 330; PsaC, Art no. AS10 939) and subsequently three 10-min washes with TBST buffer. The membrane was then probed with alkaline

phosphatase-conjugated goat anti-rabbit IgG or anti-chicken IgY for 1 h or 2 h, respectively, and washed three times again. Following three rinses with PhoA buffer (20 mM Tris, 100 mM NaCl, and 10 mM MgCl$_2$, pH 9.5), protein bands on the membrane were visualized with NBT/BCIP (Roche, Indianapolis, IN, USA) and quantified by densitometry.

Ferredoxin was also estimated by assuming 1:1 ratio to PsaC. The Fe content of the nitrogen complex was estimated as 38 per complex by assuming 2 Fe-protein dimers and 1 MoFe-protein tetramer per nitrogenase complex, 4 Fe per Fe-protein dimer, and 30 Fe per MoFe-protein tetramer. Further with the ratio of NifH to nitrogenase complex of 1:4, Fe in the nitrogenase complex [μmol Fe (g protein)$^{-1}$] was estimated to be 38/4 Fe (NifH)$^{-1}$ (Supplementary Table 3). 3, 6, 12, and 2 Fe atoms were assumed for each unit of PSII, cyt b6/f complex, PSI and ferredoxin, respectively[4], to calculate the Fe in these proteins [μmol Fe (g protein)$^{-1}$], and the sum of Fe in these proteins gave an estimate of Fe in photosystems (Supplementary Table 3). Finally, by assuming proteins accounting for 30% of dry cell mass and C for 50% of dry cell mass[4], a factor of 0.6 g protein (g C)$^{-1}$, or 7.2 g protein (mol C)$^{-1}$, was applied to convert the estimates to Fe quota [μmol Fe (mol C)$^{-1}$] (Supplementary Table 3).

**Cellular model equations**. Model parameterization schemes based on the quantitative analyses were integrated to a model only depending on two unknown parameters $f_{Fe}^{PS}$ (fraction of metabolic Fe allocated to photosystems) and $f_E^{CF}$ (fraction of produced energy allocated to C fixation) under given $Q_{Fe}$, pH and $pCO_2$:

$$g = G\big(f_{Fe}^{PS}, f_E^{CF}, Q_{Fe}, pH, pCO_2\big) \qquad (1)$$

Here we briefly describe the model equations. Description and values of all the model parameters will not be included here but are listed in Table 3.

The threshold of $Q_{Fe}$ for Fe storage is:

$$Q_{Fe}^c = Q_{Fe,bsl}^c \cdot (1 + OA^{ST} \cdot d_r H) \qquad (2)$$

where $d_r H = 10^{-(pH-pH_{bsl})} - 1$ is the relative change of medium H$^+$ concentration to a baseline condition $(pH_{bsl})$. Metabolic Fe, $Q_{Fe}^*$, is then calculated from from $Q_{Fe}^c$:

$$Q_{Fe}^* = \begin{cases} Q_{Fe}, & Q_{Fe} \leq Q_{Fe}^c \\ Q_{Fe}^c + (1 - f^{ST}) \cdot (Q_{Fe} - Q_{Fe}^c), & Q_{Fe} > Q_{Fe}^c \end{cases} \qquad (3)$$

$Q_{Fe}^{PS} = f_{Fe}^{PS} \cdot Q_{Fe}^*$ is used to determine energy production rate:

$$E = E_{max} \cdot \frac{Q_{Fe}^{PS}}{K_{Fe}^{PS} + Q_{Fe}^{PS}} \quad (4)$$

Therefore, energy allocated to C fixation, $E^{CF} = f_E^{CF} \cdot E$, is used to determined the specific C fixation rate ($g_C$) from the energy consumption rate of carbon fixation:

$$g_C = E^{CF}/ec^{CF} \quad (5)$$

Energy consumptions for several processes then can be estimated based on $g_C$. The energy consumption for the CCM is:

$$E^{CCM} = ec^{CCM} \cdot g_C \cdot \frac{1 - (1 - f^{BC}) \cdot pCO_2/pCO_{2,bsl}}{1 - l_k \cdot [1 - (pCO_2/pCO_{2,bsl} - 1) \cdot OA^{lk}]} \quad (6)$$

The energy consumption for anti-stress is:

$$E^{AtS} = max(0, ec^{AtS} \cdot g_C \cdot d_r H) \quad (7)$$

The energy consumption for maintenance is:

$$E^{MT} = ec^{MT} \cdot g_C \quad (8)$$

Thus, the residual energy is subsequently allocated to $N_2$ fixation:

$$E^{NF} = E - E^{CF} - E^{CCM} - E^{AtS} - E^{MT} \quad (9)$$

Then we can estimate the energy-limiting $N_2$ fixation rate from energy consumption rate of $N_2$ fixation:

$$NF^E = max(0, E^{NF}/ec^{NF}) \quad (10)$$

Maintenance Fe can also be estimated from $g_C$:

$$Q_{Fe}^{MT} = g_C/IUE^{MT} \quad (11)$$

Hence the residual Fe is subsequently allocated to $N_2$ fixation:

$$Q_{Fe}^{NF} = max(0, Q_{Fe}^* - Q_{Fe}^{PS} - Q_{Fe}^{MT}) \quad (12)$$

and the Fe-limiting $N_2$ fixation rate can be estimated by the following two equations accounting for effect of $Q_{Fe}^{NF}$ and pH, respectively:

$$NF^{bsl} = NF_{max} \cdot \frac{Q_{Fe}^{NF}}{K_{Fe}^{NF} + Q_{Fe}^{NF}} \quad (13)$$

$$NF^{Fe} = NF^{bsl} \cdot \left(1 - (1 - 10^{pH - pH_{bsl}}) \cdot \frac{Q_{Fe,ref}^{NF}}{Q_{Fe}^{NF}}\right) \quad (14)$$

Therefore, the modeled $N_2$ fixation rate is the smaller of $NF^E$ and $NF^{Fe}$. The N-based growth rate $g_N$ can be calculated by multiplying the $N_2$ fixation rate with a molar C:N of 5.6, an average value obtained from our culture experiments. Then the modeled *Trichodesmium* growth rate, $g$, is the smaller of $g_C$ and $g_N$.

MATLAB optimization function fminsearch[48] is used to solve Eq. (1) by optimizing the two unknown parameters $f_{Fe}^{PS}$ and $f_E^{CF}$ to obtain maximal $g$.

**Cellular model simulation.** To compare with the observations, the model first runs at the same pH and $pCO_2$ levels as the ambient and acidified conditions of the culture experiments. The minimum intracellular Fe level of the simulation is 20 μmol Fe (mol C)$^{-1}$, as from our culture experiments the diazotroph cannot survive below this level. The model is then simulated for intracellular Fe level of 20–300 μmol (mol C)$^{-1}$ and environmental pH of 7.75–8.20, with $pCO_2$ determined from varying pH and a constant alkalinity of 2200 μmol kg$^{-1}$ under salinity of 35 PSU and temperature of 35 °C using the CO2SYS program (http://cdiac.ornl.gov/ftp/co2sys/)[49].

**Model experiments of individual OA effect.** In each experiment, the model only enabled one OA effect: the reduced CCM energy consumption, the anti-stress energy consumption under lower pH or the reduced nitrogenase efficiency. Each experiment run over a $Q_{Fe}$ range of 20–300 μmol Fe (mol C)$^{-1}$ at two pH levels of 8.02 and 7.82, with the $pCO_2$ determined in same way as used above. The relative change of the modeled growth rate at the two pH levels was calculated for each experiment. The relative changes of Fe in nitrogenase and photosystems and energy production rate were also calculated for the anti-stress energy consumption only and the reduced nitrogenase efficiency only experiments.

A light-limiting experiment was set up in which the maximum energy production rate is 560 kJ (mol C)$^{-1}$ d$^{-1}$, a rate that corresponds to the ambient high-Fe treatment of the culture experiment, and the anti-stress energy consumption was enabled as the only OA effect.

**Projection of *Trichodesmium*'s potential to fix $N_2$.** Our study projects *Trichodesmium*'s potential to fix $N_2$ (per *Trichodesmium* C biomass per unit time) solely determined by Fe and pH/$pCO_2$. We first collected in situ *Trichodesmium* Fe quota ($Q_{Fe}$) measurements from literature[24,50–55], and binned them according to their sampling location (1° × 1°) and time (month) (Supplementary Figure 5). We obtained the inorganic Fe concentration (Fe′) in surface ocean in 1990s from

model CESM-BGC[56], which appears one of the best models in fitting to the measured dissolved Fe[57]. The monthly climatology of Fe′ in 1990s was calculated. $Q_{Fe}$ and the modeled Fe′ at same location/time were compared. Similar to the observation in laboratory experiments[4], a linear regression was calculated for $Log(Q_{Fe})$ and $Log(Fe')$, in which a data source[54] was excluded because it used a different method from and obtained much lower $Q_{Fe}$ than other studies, and therefore may not be directly comparable to other data sources. The obtained regression between $Log(Q_{Fe})$ and $Log(Fe')$ was then used to estimate *Trichodesmium* $Q_{Fe}$ from the CESM-BGC surface Fe′ in 1990s between 35°S and 35°N, a probable range that *Trichodesmium* may exist[12,58].

We further obtained the CESM-BGC RCP 8.5 modeled surface pH and $pCO_2$ in 1990s and 2090s (Supplementary Figure 6), which, together with the above estimated *Trichodesmium* $Q_{Fe}$, were used as inputs of the cellular model to project the monthly *Trichodesmium* $N_2$ fixation potential in these two periods. The monthly projections were averaged to annual estimates. We did not consider the change of Fe′, and therefore $Q_{Fe}$, in these two periods, because large uncertainties exist in projecting the change of the Fe deposition to the future ocean[59].

The source of the CESM-BGC data was the University Corporation for Atmospheric Research (UCAR) at http://www.earthsystemgrid.org. © 2002 University Corporation for Atmospheric Research. All Rights Reserved.

## Data availability

All measurement data generated or analyzed during this study are included in this published article and its supplementary information files. Data generated by models during this study are available from the correspondence authors on request.

## Code availability

The computer code of the model used in this study are available from the correspondence authors on reasonable request.

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

## Acknowledgements

This work was funded by the National Key R&D Program of China (2016YFA0601404 and 2016YFA0601203), NSFC (41476093, 41721005, 41890802, 31861143022 and 41376116), and the MEL internal research fund (MELRI1502).

## Author contributions

Y-W.L., D.S., and H.H. originated concept for the study. D.S., H.H., R.S., and F.Z. designed and performed laboratory experiments. Y-W.L. and D.S. designed numerical model. Y-W.L. coded and performed numerical modeling. Y-W.L., D.S., S.A.K., and B.M.H. analyzed results, improved the numerical model, and wrote the manuscript.

## Additional information

**Competing interests:** The authors declare no competing interests.

