## [Peer Review File · Nature Communications]

Reviewers' comments:

Reviewer #1 (Remarks to the Author):

The authors use a combination of empirical data and modeling to attempt to address the response of *Trichodesmium* to low iron and ocean acidification. The topic is very relevant to the community and the Nature readership at large. The authors start by assuming iron storage is zero from cells grown in low Fe, OA conditions. They report measured iron quotas and quantified components of N fixation and light reaction pathways. With some assumptions regarding the ratios of measured to unmeasured Fe containing proteins in each pathway, they calculate the maintenance iron from the difference between the measured iron quota and the sum of Fe bound in N fixation and light reaction enzymes. As written (lines 392-399), I interpreted that the quotas are calculated from the relationship observed in an earlier publication (Shi et al.) while the specific growth rates are derived from expected growth rates at a given Fe total level from another publication (Hong et al.). If that interpretation is correct (I rather hope I am wrong), I suspect some conclusions of the model may be driven by hidden differences between the behaviours of earlier cultures and those used in these experiments (elaborated on below).

There are some profound implications of the model results, including the apparent high capacity for iron storage in cells that are growing under iron limitation (lines 120-125). For example, at 70% of maximum growth rate under ambient pH, a growth-rate based interpolation of the data in table 1 suggests a full 60% of the observed cellular iron quota is not in the metabolism pool but stored (using the authors' calculation storage = total minus metabolism). Storing iron rather than investing it in iron containing, cell division-promoting, catalysts seems maladaptive and counter to the concept of optimizing growth with available resources. While the suggested sub-optimal storage may in fact be occurring (which would be to my surprise), there is no direct evidence to equivocally support this interpretation. To me, this begs the question whether there may be materially incorrect assumptions in the model and/or that the assumed growth rates and quotas differ from the quotas in the cultures used for nitrogenase and photosystem quantification.

The authors explore the sensitivity of the difference in energetic cost to the CCM for $f(BC)$, the fraction of carbon demand met by bicarbonate transport. Why was 80% used for $f(BC)$ when, if I am not mistaken, Kranz and co-authors have reported 90% in earlier publications? Also, one could expect the true value for $f(BC)$ to be lower at high pCO_2 and also lower under low Fe conditions (in both cases, the fraction of CO_2 demand for growth met by CO_2 diffusion would increase). Is there a rationale for using a constant $f(BC)$ among all treatments? As an aside, does the, "reduce to zero" (Line 197) mean there is a 100% decline in the CCM energy cost with a doubling of pCO_2 when $f(BC)$ is 50%? Can it be zero?

On two bases, I had difficulty understanding the logic used to calculate the maintenance quotas of treatments other than low iron / OA condition. The authors used low iron / OA condition maintenance quota and growth rates of other treatments to calculate maintenance quotas of other treatments (OA and ambient). This maintenance quota includes cellular iron used for respiration, OA stress mitigation and any other parameters not quantified by N fixation or photosynthesis (or storage for low Fe/OA). Some of this maintenance is likely running H^+ efflux pumps, suggesting homeostatic energy demand is higher for OA cells – and this would be decoupled from growth rate. Second, if cytosolic pH is lower with OA (as suggested by the authors' earlier work; Hong et al 2017), any decreases in catalytic rate for other (ie not N fixation and photosynthesis) Fe requiring processes would lead to higher maintenance costs (thus, the growth rate conversion of maintenance quota among treatments would be inappropriate). Relatedly, if it is fair to assume the

stress response due to OA involves Fe-containing catalysts (supported by the authors' explicit inclusion of an energy demand for this stress response and that iron is involved in many redox processes), this would further contribute to a difference in maintenance iron demand between OA and contemporary conditions. All this said, it's only fair to point out that, if the maintenance quotas are truly as low as reported (requiring other model assumptions are not materially correct), this point would probably not have any meaningful effect on the conclusions.

L316-319: This argument needs to be clarified. It seems the authors are suggesting iron limited cells would save on CCM energy because the passive flux satisfies a higher portion of the C demand. That is fine but it seems the authors argued earlier that this is negligible. More importantly, doesn't optimization suggest only the correct amount of iron would be allocated to nitrogenase (rather than an imbalanced scenario that leads to a detrimentally lower energy production by photosynthesis)?

The N fixation results are estimated from a fixed stoichiometry of acetylene reduction to ammonium production among treatments where this ratio may reasonably be expected to differ by as much as 33%. Also, the authors discuss the production of H₂ at the expense of ammonium production with reduced energy supply, suggesting this 33% difference may be at play. How does this influence the results presented?

Line 359: Nitrogenase is an old enzyme, yes, but oceanic pH has been at least as low as the levels tested in recent geologic time. The assertion that *Trichodesmium* may not be able to survive anthropogenic OA (Line 303) seems unsupported.

Minor points

Most readers do now know what RCP 8.5 means; avoid jargon in the abstract.

L19: Cyanobacterium is singular, whereas *Trichodesmium* is a genus of several cyanobacteria.

L43-44, 49: Awkward wording.

Line 173: The authors write, "The CO₂ passively diffusing into cytoplasm is converted to HCO₃ at the thylakoid membrane, which also costs energy by oxidizing NADPH". Where is the redox reaction here, and why can't this process be accomplished by carbonic anhydrase?

Reviewer #2 (Remarks to the Author):

This well written manuscript presents a new optimality-based model for the nitrogen fixing phytoplankton *Trichodesmium* and its application, in combination with simulation results from an independent Earth System climate model, to predict how oceanic nitrogen fixation will respond under a future scenario of ocean acidification. The quantitative measurements of different functional pools of intracellular iron are a strong feature of this work in that they provide a sound basis for the model formulation. I find the study well designed and executed. This topic is timely, and the results presented are of interest to many researchers and readers of this journal, because of their implications for the response of marine ecosystems and biogeochemical cycles to ocean acidification and climate change.

I recommend publication, provided the authors respond to a few minor concerns below to the satisfaction of the editors.

Specific comments:

lines 124-125: "Trichodesmium put aside large amount of Fe, even if its growth was in need of more Fe."

This is important in the context of the optimality argument underlying the model used herein, and indeed many recent models. It implies that if these organisms are indeed optimally allocating their

resources, they are doing with respect to a longer time-scale than the immediate sense in which most of these models calculate optimal strategies. In other words, why not optimize for growth immediately? Probably because under natural fluctuations in iron availability it is advantageous to save some Fe for later use.

I understand that the authors have chosen to formulate their model response based on the observed response, which I agree is wise. For basic understanding and further model development, however, I think it would be helpful if the authors could comment on and at least to some extent clarify their views concerning the timescale for optimal resource allocation. This is particularly desirable given their concluding remarks about the kind of experiments and approaches needed to further clarify how ocean acidification may affect nitrogen fixation.

The model formulation seems sound, and well grounded in observations, except for one point which could use more justification:

lines 179-181: On what basis do the authors assume that this energetic cost adds 20% to cost HCO₃⁻ transport?

Some justification, or at least an assessment of the relative importance of this assumed cost, is needed. To what degree does this assumption impact the conclusions of this study?

Fig. 3. The final two panels are mislabeled (lettering) in the caption. They should be E and F as labeled in the figure.

Reviewer #3 (Remarks to the Author):

This manuscript builds on previous experimental work of Hong and Shi to develop a mechanistic cellular model of N₂ fixation in *Trichodesmium*. Using the model, the authors are able to separate the effects of OA and Fe limitation, and to examine the interactions. I found this paper to be exciting, since it provides a simplified mechanistic understanding of how *Tricho* would respond to changing pH. The manuscript is generally well-written, and the logic behind the model seems robust. As with any model of this sort some assumptions have to be made, but those are clearly laid out and reasonable.

My biggest criticism of the model itself is that there are relatively little data to parameterize/test the model against. For example, Figure 2 and 3, where linear or Monod fits are applied to (often) two data points. This inherently assumes that the underlying relationship is known, and doesn't allow more complex relationships. This is not going to be resolved for this manuscript (I'm not expecting the authors to generate a more complete lab-based dataset) but I think they should discuss a bit the inherent limitations of the data.

Related to that, their conclusions are in direct contrast to several published papers that suggest OA will enhance N₂ fixation (lines 325-329). While I agree that, based on their model, it's unlikely, they don't really address the discrepancy between the model results and these publications. If it's not being caused by changes in the CCM, then what would explain these discrepancies? This should really be expanded upon, even if it's simply to reiterate the conclusions of Hong et al.

Finally, while they mention that there are other factors not included (temperature, P, etc.) it would be useful to include a more complete discussion of the covarying effects of temperature and OA, since those are the big drivers under RCP 8.5. For example, Boatman et al. (2017) concluded that changing temperature will expand the thermal niche of *Tricho*, which would certainly modulate the data presented in Figure 5. I understand that's not the main point of this paper but it should still be mentioned.

Some specific comments are below.

Line 104: resolve, not revolve

Lines 123-128: my understanding is that Table 1 is based on equilibrium quotas, so it's not really "luxury uptake", which refers to active transport (uptake) of a substrate when that substrate is no longer rate-limiting for (e.g.) growth. The fact that the Fe was being stored rather than used, even though "growth was in need of more Fe" sounds more like a description of dynamic allocation rather than luxury uptake. That leads to the question of whether the values in Table 1 are really from equilibrium conditions, which (I think) would be required to conduct the mass-balance partitioning of Fe within the cell that is being used.

Lines 140-142: were the proton concentrations determined directly in this study? Using the phrase "as observed previously" suggests that new data were generated, but the Methods say that the Hong et al. dataset was used. It would be good to be more clear here (lines 140-142) that no new data were generated.

Line 149: I understand that generating these data is a LOT of work, but fitting a Monod curve to two points is somewhat questionable. I don't really expect the authors to redo a bunch of culture work, but this is definitely a limitation of the analysis, even with the very tight error bars on the two points.

Figure 2C: should be anti-stress?

Figure 3: panel F (percent change) is not described in the figure legend

Figure S6: should be ppmv, correct?

Line 299: constraints, not constrains

Note our response to Reviewers' comments are in blue throughout this document.

Reviewer #1 (Remarks to the Author):

The authors use a combination of empirical data and modeling to attempt to address the response of *Trichodesmium* to low iron and ocean acidification. The topic is very relevant to the community and the Nature readership at large. The authors start by assuming iron storage is zero from cells grown in low Fe, OA conditions. They report measured iron quotas and quantified components of N fixation and light reaction pathways. With some assumptions regarding the ratios of measured to unmeasured Fe containing proteins in each pathway, they calculate the maintenance iron from the difference between the measured iron quota and the sum of Fe bound in N fixation and light reaction enzymes.

We are glad that the Reviewers found the topic of our work very relevant to the community and the Nature readership at large.

As written (lines 392-399), I interpreted that the quotas are calculated from the relationship observed in an earlier publication (Shi et al.) while the specific growth rates are derived from expected growth rates at a given Fe total level from another publication (Hong et al.). If that interpretation is correct (I rather hope I am wrong), I suspect some conclusions of the model may be driven by hidden differences between the behaviours of earlier cultures and those used in these experiments (elaborated on below).

There are some profound implications of the model results, including the apparent high capacity for iron storage in cells that are growing under iron limitation (lines 120-125). For example, at 70% of maximum growth rate under ambient pH, a growth-rate based interpolation of the data in table 1 suggests a full 60% of the observed cellular iron quota is not in the metabolism pool but stored (using the authors' calculation storage = total minus metabolism). Storing iron rather than investing it in iron containing, cell division-promoting, catalysts seems maladaptive and counter to the concept of optimizing growth with available resources. While the suggested sub-optimal storage may in fact be occurring (which would be to my surprise), there is no direct evidence to equivocally support this interpretation. To me, this begs the question whether there may be materially incorrect assumptions in the model and/or that the assumed growth rates and quotas differ from the quotas in the cultures used for nitrogenase and photosystem quantification.

As the comments in the previous two paragraphs are related, here we respond them together.

The Reviewer was correct in that the intracellular Fe quota (Q_{Fe}) was estimated from inorganic Fe concentration in medium (Fe') using the relationship between Fe' and Q_{Fe} observed in Shi et al. [2012], while nitrogenase and photosynthetic Fe-containing proteins were experimentally quantified using samples from Hong et al. [2017]. The Reviewer thus suspected that the possible difference between experiments in the two publications would have led to incorrect estimation for Q_{Fe} and thus the unexpectedly high estimate of storage Fe. We understand the Reviewer's concern and thank the Reviewer for reminding us to clarify this.

(1) We understand that the Reviewer was particularly concerned about the estimated high percentage of storage Fe under high Fe conditions. In order to exclude the possibility that the high estimated storage Fe was caused by using the data obtained with two different studies, we estimated Fe storage using the data of Shi et al. [2012] to see whether the result would be different from that obtained with Hong et al. [2017] data following the same method of the present study.

Table R1. Expression of photosynthetic proteins and nitrogenase and the estimated Fe quota in these proteins of *Trichodesmium* grown at high Fe in Shi et al. [2012] and Hong et al. [2017].

	Shi et al. [2012]				Hong et al. [2017]			
	Protein content [μmol (g total protein) ⁻¹]		Fe quota [$\mu\text{mol Fe (mol C)}^{-1}$]		Protein content [μmol (g total protein) ⁻¹]		Fe quota [$\mu\text{mol Fe (mol C)}^{-1}$]	
	Ambient	Acidified	Ambient	Acidified	Ambient	Acidified	Ambient	Acidified
NifH	0.525±0.009	0.653±0.030	35.9±0.6	44.6±2.0	0.552±0.067	0.638±0.050	37.8±4.5	43.6±3.4
PsbA	0.118±0.004	0.113±0.009	2.55±0.09	2.44±0.19	0.083±0.010	0.084±0.005	1.79±0.21	1.82±0.11
PetC	-	-	-	-	0.033±0.000	0.047±0.008	1.42±0.02	2.04±0.34
PsaC	0.051±0.023	0.071±0.012	4.43±2.00	6.15±1.06	0.097±0.001	0.128±0.009	8.41±0.09	11.03±0.76

Table R1 shows the expression of key photosynthetic proteins and NifH as well as the corresponding Q_{Fe} in the proteins calculated based on protein quantification in both of the studies under high Fe conditions. We can see that the abundance of PsbA and PsaC observed in the two studies agree well with each other. Although PetC was not measured in Shi et al. [2012], it only contributes to a minor fraction of the intracellular Fe pool. Thereby the two studies observe similar Q_{Fe} in the photosystems. The expression of NifH under high Fe conditions is also very close between the two studies. Therefore, the expression of these key Fe-containing proteins is consistent in the two studies, further suggesting that the experimental results of the two studies are reproducible.

We further calculated the Fe storage at high Fe (Table R2), indicating that Q_{Fe} in nitrogenase and photosystems are only representing ~10% of the total Q_{Fe} , with the remaining ~90% of Fe in maintenance and storage (among which the maintenance Fe is a minor fraction; see below for our responses to the comments on maintenance Fe). Therefore, in Shi et al. [2012] the estimated storage Fe at high Fe is also significant (up to ~90%) and is as high as that estimated using data of Hong et al. [2017] in the current manuscript.

Table R2. Assessment of Fe distribution ($\mu\text{mol Fe (mol C)}^{-1}$) in different intracellular pools in *Trichodesmium* grown at high Fe using data of Shi et al. [2012] and Hong et al. [2017] for comparison.

	Shi et al. [2012]		Hong et al. [2017]	
	Ambient	Acidified	Ambient	Acidified
Total intracellular	411	411	328	331
Nitrogenase	35.9	44.6	37.8	43.6
Photosystems	8.5	10.6	13.0	16.7
Storage & Maintenance	367	356	277	271
% Storage & Maintenance	89%	87%	84%	82%

(2) The experimental conditions in Hong et al. [2017], including the *Trichodesmium* strain, culture medium, light intensity and temperature, etc. are the same as those in Shi et al. [2012]. Despite the fact that two studies were conducted several years apart in two different laboratories, the observed growth rates and particularly its response to OA under both Fe-limited and Fe-replete conditions are largely reproducible (see Table R3 below). In addition, Fig. 2A in Shi et al. [2012] clearly demonstrates that the strong correlation between Q_{Fe} and Fe' is independent of pH, $p\text{CO}_2$, and growth rate. Therefore, we are very confident that the Q_{Fe} -Fe' relationship observed in Shi et al. [2012] can be applied to the same organism grown under the same experimental conditions in Hong et al. [2017]. We have now clarified and included this information in the revised manuscript (**Lines 438-440**).

Table R3. Growth rates of Fe-limited and Fe-replete *Trichodesmium* under ambient and acidified conditions and its response to OA observed in Shi et al. [2012] and Hong et al. [2017].

	Shi et al. [2012]			Hong et al. [2017]		
	Growth rate			Growth rate		
	Ambient	Acidified	% change	Ambient	Acidified	% change
Fe-limited	0.26 ± 0.02	0.19 ± 0.01	-27	0.31 ± 0.01	0.22 ± 0.02	-29
Fe-replete	0.46 ± 0.01	0.37 ± 0.01	-20	0.56 ± 0.02	0.46 ± 0.02	-18

(3) Intracellular Fe in *Trichodesmium* can be stored in both Dps [Castruita et al., 2006] and ferritin [Harrison and Arosio, 1996]. Here we estimated the maximum Fe storage capacity of Dps and ferritin. As antibodies and standards for Dps and ferritin (frit) are not commercially available for absolute protein quantification, we estimated their amounts based on the numbers of unique spectra obtained by mass spectrometry, assuming that the spectra numbers are proportional to the protein/peptide amounts [Lal et al., 2009; Zeng et al., 2018; Zhang et al., 2015]. PsaC as has been quantified by western blot was used as a reference. The Dps and frit

subunits were roughly 79% and 3 times of PsaC, respectively (Supplementary Table 1). As each Dps protein contains 12 subunits and each frt contains 24 subunits, the abundance of Dps and frt were 6.7% and 12.5% of PsaC, respectively (Supplementary Table 1). Given each Dps protein can bind up to 260 Fe atoms [Castruita *et al.*, 2006] and each frt can bind up to 4500 Fe [Harrison and Arosio, 1996], using the measured Fe quota in PsaC (Supplementary Table 3), maximum Fe quota in Dps and frt can be calculated as:

$$Q_{\text{Fe}}^{\text{Dps}} = Q_{\text{Fe}}^{\text{PsaC}} \cdot \frac{n_{\text{Fe}}^{\text{Dps}}}{n_{\text{Fe}}^{\text{PsaC}}} \cdot r_{\text{PsaC}}^{\text{Dps}},$$

$$Q_{\text{Fe}}^{\text{frt}} = Q_{\text{Fe}}^{\text{PsaC}} \cdot \frac{n_{\text{Fe}}^{\text{frt}}}{n_{\text{Fe}}^{\text{PsaC}}} \cdot r_{\text{PsaC}}^{\text{frt}},$$

where $Q_{\text{Fe}}^{\text{Dps}}$ and $Q_{\text{Fe}}^{\text{PsaC}}$ are maximum Fe quotas in Dps and PsaC, respectively, $n_{\text{Fe}}^{\text{PsaC}} = 12$ is the number of Fe atoms per PsaC, $n_{\text{Fe}}^{\text{Dps}} = 260$ is the maximum number of Fe atoms per Dps, $n_{\text{Fe}}^{\text{frt}} = 4500$ is the maximum number of Fe atoms per frt, and $r_{\text{PsaC}}^{\text{Dps}}$ and $r_{\text{PsaC}}^{\text{frt}}$ are ratios of Dps and frt abundance to PsaC abundance, respectively, based on the spectrum analysis.

The estimated maximum Fe storage capacity of Dps and frt are 406 and 533 $\mu\text{mol Fe (mol C)}^{-1}$, respectively, for Fe-replete *Trichodesmium* under ambient and acidified conditions (Supplementary Table 2), which are much higher than our estimated Fe storage (Table 1). While it is unlikely that all Dps or frt has its maximum Fe bound, our analysis suggests that a high Fe storage as estimated (Table 1) is feasible in *Trichodesmium*. However, it remains unclear why the diazotroph would store such a high amount of Fe, which is certainly worthy of further investigation. We have now revised the manuscript (**Lines 133-137**) and also included this information in the supplementary information.

The authors explore the sensitivity of the difference in energetic cost to the CCM for $f(\text{BC})$, the fraction of carbon demand met by bicarbonate transport. Why was 80% used for $f(\text{BC})$ when, if I am not mistaken, Kranz and co-authors have reported 90% in earlier publications? Also, one could expect the true value for $f(\text{BC})$ to be lower at high $p\text{CO}_2$ and also lower under low Fe conditions (in both cases, the fraction of CO_2 demand for growth met by CO_2 diffusion would increase). Is there a rationale for using a constant $f(\text{BC})$ among all treatments? As an aside, does the, “reduce to zero” (Line 197) mean there is a 100% decline in the CCM energy cost with a doubling of $p\text{CO}_2$ when $f(\text{BC})$ is 50%? Can it be zero?

First to clarify, f^{BC} was a parameter of the contribution of bicarbonate to total inorganic carbon transport under **baseline** (i.e., **ambient**) conditions (Table 3). The effective contribution of bicarbonate did vary with CO_2 in our model because of the changes in CO_2 diffusion and leakage, which is controlled by Eq. (6). This was already described and discussed in the original manuscript (**Lines 198-208**). [Note that we did incorrectly mark the Equation number in the original manuscript, and have now changed it to “Eq. (6)”].

Please note that the fraction of $>90\%$ (or even $>100\%$) reported in Kranz *et al.* [2009] is for bicarbonate uptake:**C fixation** (as estimated using the ^{14}C disequilibrium method in Kranz *et al.* [2009]), whereas f^{BC} in our model is bicarbonate uptake:**C uptake** [i.e., bicarbonate uptake:(C

fixation + C leakage)]. Thus f^{BC} is lower in our study. In fact, the combined usage of $f^{BC} = 80\%$ and C leakage $l_k = 50\%$, as used in our model, is suggested by a paper co-authored by Kranz and coauthors [Eichner *et al.*, 2014], which has now been cited in the revised manuscript.

We agree with the Reviewer that f^{BC} could be smaller under a lower growth rate as a larger fraction of C uptake can be met by CO₂ diffusion. We nevertheless used a constant f^{BC} in the model, for (1) simplify of model construction and (2) CCM energy is a small fraction in total energy consumption and thus changes in f^{BC} does not change model results substantially. We have now included this discussion in the manuscript (**Lines 208-210, Lines 213-214**). Moreover, we also tested the model sensitivity to f^{BC} and l_k (**Lines 210-213**, and Supplementary Fig. 2), which reveals that the estimated CCM energy cost can vary greatly with f^{BC} and less with l_k .

The statement that the CCM energy cost could be reduced to zero is only a hypothetical consideration as part of the sensitivity test. In the test if f^{BC} is $< 50\%$ (and so more than 50% of inorganic carbon demand is met by CO₂ diffusion) a doubling of CO₂ allows CO₂ uptake (which does not cost energy) to fully meet cellular demands hence eliminating CCM energy consumption. However, as f^{BC} is often found to be higher than 70-80% in cyanobacteria, a doubling of $p\text{CO}_2$, would not significantly decrease bicarbonate transport, neither the overall CCM energy cost. In addition, this “zero” energy is under the framework of our model: there must be other energy needed for CCM; in this study we only consider energy used in bicarbonate transport and neglect other energy consumption (which we believe is very low compared to energy by bicarbonate transport, see [Hopkinson *et al.*, 2011]). To clarify, we have now revised the sentence (**Lines 210-213**).

On two bases, I had difficulty understanding the logic used to calculate the maintenance quotas of treatments other than low iron / OA condition. The authors used low iron / OA condition maintenance quota and growth rates of other treatments to calculate maintenance quotas of other treatments (OA and ambient). This maintenance quota includes cellular iron used for respiration, OA stress mitigation and any other parameters not quantified by N fixation or photosynthesis (or storage for low Fe/OA). Some of this maintenance is likely running H⁺ efflux pumps, suggesting homeostatic energy demand is higher for OA cells – and this would be decoupled from growth rate. Second, if cytosolic pH is lower with OA (as suggested by the authors’ earlier work; Hong *et al* 2017), any decreases in catalytic rate for other (ie not N fixation and photosynthesis) Fe requiring processes would lead to higher maintenance costs (thus, the growth rate conversion of maintenance quota among treatments would be inappropriate). Relatedly, if it is fair to assume the stress response due to OA involves Fe-containing catalysts (supported by the authors’ explicit inclusion of an energy demand for this stress response and that iron is involved in many redox processes), this would further contribute to a difference in maintenance iron demand between OA and contemporary conditions. All this said, it’s only fair to point out that, if the maintenance quotas are truly as low as reported (requiring other model assumptions are not materially correct), this point would probably not have any meaningful effect on the conclusions.

The maintenance Fe indeed includes cellular Fe involved in a variety of metabolic pathways. To simplify it for model construction and simulation, we used the maintenance Fe estimated for the low-Fe acidified treatment as a baseline and then calculated the maintenance Fe for other

treatments using growth rates. We agree with the Reviewer that the maintenance Fe is likely higher for *Trichodesmium* under OA, and we also understand the Reviewer's main concerns: 1) the maintenance Fe may be decoupled from growth rates, and 2) whether our model may have underestimated the maintenance Fe of the three conditions other than low-Fe OA.

Table R4 below summarizes the relative expression (normalized to low-Fe acidified condition) of all the Fe-containing proteins that are known or have been annotated in the Swissprot database (except those in photosynthesis, nitrogenase, Dps, and ferritin) under low-Fe ambient, high-Fe acidified and high-Fe ambient conditions (This unpublished proteomic dataset is presented in a manuscript currently in preparation for submission to another journal). These Fe-containing proteins likely contribute to maintenance Fe quota. As we can see:

Under low-Fe conditions, the expression levels of the Fe-containing proteins are on average 20% higher under ambient conditions than under acidified conditions, yet the growth rate under ambient conditions is 41% higher than that under acidified conditions. Thus, we believe that we probably may have overestimated the maintenance Fe for the low Fe-ambient treatment. Here we did not take into account the number of Fe bound by each Fe-protein, as many of them have not been well characterized and the exact Fe content is unknown. Among the 38 proteins, there are only 8 proteins (in red) that are expressed at slightly higher levels (1.45-2.12) than the relative ratio of growth rate (1.41), when comparing ambient to acidified conditions. In addition, based on the numbers of unique spectra (which can be used to estimate protein abundance [*Lal et al.*, 2009; *Zeng et al.*, 2018; *Zhang et al.*, 2015]), the 8 proteins (125 unique spectra in total) only contribute to a small fraction of the total maintenance protein pool (761 unique spectra in total). Among the 8 proteins, the abundance of 7 proteins is lower than that of the reference protein PsaC, with only 1 protein being somewhat more abundant than PsaC. Therefore, these 8 low abundant proteins should not significantly increase the maintenance quota when they are only expressed at slightly higher levels than the relative ratio of growth rate.

Under high-Fe conditions, the average Fe-containing protein expression were 44% and 49% higher under acidified and ambient conditions, respectively, than the low Fe-acidified treatment; in contrast, the relative growth rates were 110% and 155% higher under acidified and ambient conditions, respectively, than the low Fe-acidified treatment. This comparison suggests that we may also have overestimated maintenance Fe under high Fe conditions by assuming it being coupled to growth rate. In addition, under high-Fe conditions, among the Fe-proteins in Table R4, there are only 4 low abundant proteins (according to their numbers of unique spectra) that are expressed at somewhat higher levels than the relative ratios of growth rate, suggesting that the maintenance Fe is unlikely to be significantly higher than we calculated.

[Redacted]

Overall, our analysis above indicates that using the maintenance Fe in low-Fe OA as baseline, we may have **overestimated**, instead of underestimated, maintenance Fe for the three treatments other than low-Fe OA. As the maintenance Fe pool was nevertheless very small in all the treatments, its estimation should not substantially impact our model results. Moreover, our analysis indicates that the calculated high Fe storage is not due to underestimation of maintenance Fe. We have now revised the manuscript accordingly (**Lines 112-119**).

L316-319: This argument needs to be clarified. It seems the authors are suggesting iron limited cells would save on CCM energy because the passive flux satisfies a higher portion of the C demand. That is fine but it seems the authors argued earlier that this is negligible. More importantly, doesn't optimization suggest only the correct amount of iron would be allocated to nitrogenase (rather than an imbalanced scenario that leads to a detrimentally lower energy production by photosynthesis)?

This section is to describe a possible scenario where the beneficial effect on growth of the energy saved from CCM downregulation is even more negligible. As shown by our assessments, downregulation of HCO_3^- uptake while increasing CO_2 utilization under high CO_2 can save only 3% of the total cellular energy requirement. Such a small saving in energy would result in an increase of growth rate by even less than 3%, as growth needs not only energy but also Fe, and reallocation of limited cellular Fe from photosystems to nitrogenase in response to OA would partially offset the energetic benefit gained from CCM downregulation. Certainly, our optimality-based model can allocate balanced amount of Fe among nitrogenase, photosystems and other pools. This section describes how the new balance is established when changing from ambient to OA conditions. We have now revised the section for clarification (**Lines 344-348**).

The N fixation results are estimated from a fixed stoichiometry of acetylene reduction to ammonium production among treatments where this ratio may reasonably be expected to differ by as much as 33%. Also, the authors discuss the production of H_2 at the expense of ammonium production with reduced energy supply, suggesting this 33% difference may be at play. How does this influence the results presented?

We agree with the reviewer that the ratio of C_2H_2 reduction: N_2 fixation could vary as seawater pH and/or Fe condition change. We used a ratio of 4:1 to convert C_2H_2 reduction to N_2 fixation in our study, but it is possible that the ratio could be lower under ambient conditions than under acidified conditions for the reason the reviewer has already pointed out. Using a fixed ratio of C_2H_2 reduction: N_2 fixation (4:1) may thus overestimate the actual N_2 fixation rates under acidified conditions relative to those under ambient conditions. As our model has been tuned to fit the lab-based N_2 fixation rates, it is possible that the model could underestimate the impact of OA on *Trichodesmium* N_2 fixation by up to 33%. We have now discussed this in the revised manuscript (**Lines 269-276**).

Line 359: Nitrogenase is an old enzyme, yes, but oceanic pH has been at least as low as the levels tested in recent geologic time. The assertion that *Trichodesmium* may not be able to survive anthropogenic OA (Line 303) seems unsupported.

We agree and have now removed this sentence.

Minor points

Most readers do now know what RCP 8.5 means; avoid jargon in the abstract.

We have now replaced “RCP 8.5” by “if anthropogenic CO₂ emissions continue to rise” in the abstract (**Lines 29-30**). We have also given full definition of RCP 8.5 in the main text where it first appeared (**Lines 313-314**).

L19: Cyanobacterium is singular, whereas Trichodesmium is a genus of several cyanobacteria.

We have now changed “cyanobacterium” to “cyanobacteria” (**Line 19**).

L43-44, 49: Awkward wording.

We have now rephrased the sentences as “The growth enhancement of marine diazotrophs under OA is often attributed to the downregulation of CO₂-concentrating mechanisms (CCM) under high CO₂ concentration, which seemingly saves energetic resources for other cellular processes including N₂ fixation” (**Lines 40-43**), and “*Trichodesmium* needed to invest additional cellular resources and energy to cope with the stress imposed by low pH (e.g., cytosolic pH disturbance)” (**Lines 46-47**).

Line 173: The authors write, “The CO₂ passively diffusing into cytoplasm is converted to HCO₃⁻ at the thylakoid membrane, which also costs energy by oxidizing NADPH”. Where is the redox reaction here, and why can't this process be accomplished by carbonic anhydrase?

The redox reaction is shown in Fig. 9 in [Price *et al.*, 2002]. We have now clarified this sentence in the revised manuscript briefly explaining the CO₂ to HCO₃⁻ conversion (**Lines 182-188**). Internal carbonic anhydrase (catalyzing the interconversion of HCO₃⁻ and CO₂) in the cytosol is absent in cyanobacteria [Price and Badger, 1989]. We have added the respective papers in the revised manuscript.

Reviewer #2 (Remarks to the Author):

This well written manuscript presents a new optimality-based model for the nitrogen fixing phytoplankton *Trichodesmium* and its application, in combination with simulation results from an independent Earth System climate model, to predict how oceanic nitrogen fixation will respond under a future scenario of ocean acidification. The quantitative measurements of different functional pools of intracellular iron are a strong feature of this work in that they provide a sound basis for the model formulation. I find the study well designed and executed. This topic is timely, and the results presented are of interest to many researchers and readers of this journal, because of their implications for the response of marine ecosystems and biogeochemical cycles to ocean acidification and climate change.

I recommend publication, provided the authors respond to a few minor concerns below to the satisfaction of the editors.

We appreciate the positive comments of the Reviewer on our study.

Specific comments:

lines 124-125: "Trichodesmium put aside large amount of Fe, even if its growth was in need of more Fe."

This is important in the context of the optimality argument underlying the model used herein, and indeed many recent models. It implies that if these organisms are indeed optimally allocating their resources, they are doing with respect to a longer time-scale than the immediate sense in which most of these models calculate optimal strategies. In other words, why not optimize for growth immediately? Probably because under natural fluctuations in iron availability it is advantageous to save some Fe for later use.

I understand that the authors have chosen to formulate their model response based on the observed response, which I agree is wise. For basic understanding and further model development, however, I think it would helpful if the authors could comment on and at least to some extent clarify their views concerning the timescale for optimal resource allocation. This is particularly desirable given their concluding remarks about the kind of experiments and approaches needed to further clarify how ocean acidification may affect nitrogen fixation.

We thank the Reviewer for this very constructive comment, which has now been incorporated into Discussion in the revised manuscript (**Lines 391-401**) as following:

“The significant luxury uptake and subsequent storage of Fe by Trichodesmium (90% of cellular Fe above a threshold in this study), when the Fe supply is high, likely reflect its response to natural fluctuations in Fe availability by storing Fe for later use. On a longer time-scale, it is possible that the diazotroph may evolve to optimally allocate its cellular Fe, investing more in metabolism than in storage to compensate for the reduced nitrogenase efficiency. This phenomenon has already been found in our study where the Fe storage started at a higher threshold of intracellular Fe (i.e., more Fe used in metabolic processes) under OA (Fig. 2A), although the full extent to which this can compensate for the negative effects of OA is unclear. If the compensation becomes increasingly significant in the future, OA effect on Trichodesmium N₂ fixation would be less pronounced than that projected in this study, particularly in areas with high Fe supply.”

The model formulation seems sound, and well grounded in observations, except for one point which could use more justification:

lines 179-181: On what basis do the authors assumed that this energetic cost adds 20% to cost HCO₃⁻ transport?

Some justification, or at least an assessment of the relative importance of this assumed cost, is needed. To what degree does this assumption impact the conclusions of this study?

We thank the Reviewer for the comment. To the best of our knowledge, there is no experimental observation supporting the assumption that building HCO_3^- transporters and carboxysomes requires additional 20% energy besides that needed for transporting HCO_3^- . Nevertheless, we conducted a model sensitivity test by increasing this extra energy cost to 100% (i.e., the energy for transporting HCO_3^- equals to that for building HCO_3^- transporters and carboxysomes), and then the CCM energy cost rate increase from 96 to 160 $\text{kJ} (\text{mol C})^{-1}$. The results show that the benefit of CCM downregulation under a doubling of $p\text{CO}_2$ increased from 0.6% to 1.0%, which is still very limited and thus does not change our conclusions.

We have now revised the manuscript to address this issue (**Lines 194-197**).

Fig. 3. The final two panels are mislabeled (lettering) in the caption. They should be E and F as labeled in the figure.

The Reviewer has a keen eye. We have now corrected the labels in the figure.

Reviewer #3 (Remarks to the Author):

This manuscript builds on previous experimental work of Hong and Shi to develop a mechanistic cellular model of N_2 fixation in *Trichodesmium*. Using the model, the authors are able to separate the effects of OA and Fe limitation, and to examine the interactions. I found this paper to be exciting, since it provides a simplified mechanistic understanding of how *Tricho* would respond to changing pH. The manuscript is generally well-written, and the logic behind the model seems robust. As with any model of this sort some assumptions have to be made, but those are clearly laid out and reasonable.

We are thankful to the Reviewer for the positive comments, and are happy that the Reviewer found our work exciting.

My biggest criticism of the model itself is that there are relatively little data to parameterize/test the model against. For example, Figure 2 and 3, where linear or Monod fits are applied to (often) two data points. This inherently assumes that the underlying relationship is known, and doesn't allow more complex relationships. This is not going to be resolved for this manuscript (I'm not expecting the authors to generate a more complete lab-based dataset) but I think they should discuss a bit the inherent limitations of the data.

We agree with the Reviewer that there are relatively little data for us to parameterize and test the model. We have now added a paragraph in the revised manuscript to discuss this limitation (**Lines 262-269**):

“It should be noted that although linear or Monod-like equations are assumed and applied in the cellular model, N_2 fixation rates and intracellular Fe allocations which the model is parameterized and tested against are derived experimentally at only two Fe levels (Fig. 2). As the

actual underlying relationships are unknown, it is possible that our model results may be off to a certain degree in particular at moderate levels of Fe. Nevertheless, the model reasonably fitted the measured growth rates observed at multiple Fe concentrations (i.e., 5 under ambient conditions and 4 under acidified conditions) (Fig. 3A), indicating that the assumed relationships in our model are reasonably sound.”

Related to that, their conclusions are in direct contrast to several published papers that suggest OA will enhance N₂ fixation (lines 325-329). While I agree that, based on their model, it’s unlikely, they don’t really address the discrepancy between the model results and these publications. If it’s not being caused by changes in the CCM, then what would explain these discrepancies? This should really be expanded upon, even if it’s simply to reiterate the conclusions of Hong et al.

A discussion on the discrepancies between our results and the previous papers has now been included in the revised manuscript (**Lines 361-364**).

Finally, while they mention that there are other factors not included (temperature, P, etc.) it would be useful to include a more complete discussion of the covarying effects of temperature and OA, since those are the big drivers under RCP 8.5. For example, Boatman et al. (2017) concluded that changing temperature will expand the thermal niche of *Tricho*, which would certainly modulate the data presented in Figure 5. I understand that’s not the main point of this paper but it should still be mentioned.

We agree with the Reviewer and have now included this in Discussion (**Lines 418-424**):

“Growth of *Trichodesmium* is often limited by the deficiency of phosphorus in surface seawater, which is expected to intensify as a result of augmented water column stratification caused by global warming. Additionally, a recent study shows that the optimal thermal range of *Trichodesmium* becomes wider under higher *p*CO₂ and thus increasing temperature can help the diazotroph to alleviate the impact of OA. Therefore, changes in these factors in the future ocean may modulate the OA effect and hence could change our model projections.”

Some specific comments are below.

Line 104: resolve, not revolve

Corrected (**Line 103**).

Lines 123-128: my understanding is that Table 1 is based on equilibrium quotas, so it’s not really “luxury uptake”, which refers to active transport (uptake) of a substrate when that substrate is no longer rate-limiting for (e.g.) growth. The fact that the Fe was being stored rather than used, even though “growth was in need of more Fe” sounds more like a description of dynamic allocation rather than luxury uptake. That leads to the question of whether the values in Table 1 are really

from equilibrium conditions, which (I think) would be required to conduct the mass-balance partitioning of Fe within the cell that is being used.

The Reviewer is right that the values in Table 1 are obtained with *Trichodesmium* grown under steady-state conditions. Therefore, we also agree with the Reviewer that the “luxury Fe uptake” we refer to in our manuscript is actually “luxury Fe storage”, but not active transport of Fe. We have now rephrased it as “Such a phenomenon is often the result of ‘luxury Fe uptake’ and is commonly observed in the field.” (Lines 129-130).

Lines 140-142: were the proton concentrations determined directly in this study? Using the phrase “as observed previously” suggests that new data were generated, but the Methods say that the Hong et al. dataset was used. It would be good to be more clear here (lines 140-142) that no new data were generated.

We thank the Reviewer for the reminder. Yes, the data are from Hong et al. 2017. We have now clarified it by deleting “as observed previously” (Line 150).

Line 149: I understand that generating these data is a LOT of work, but fitting a Monod curve to two points is somewhat questionable. I don’t really expect the authors to redo a bunch of culture work, but this is definitely a limitation of the analysis, even with the very tight error bars on the two points.

We are thankful to the Reviewer for appreciating our efforts in carrying out the study and obtaining the data. We do agree with the Reviewer that there are relatively little data for us to parameterize and test the model. We have now discussed this inherent limitation in the revised manuscript to (Lines 262-269):

“It should be noted that although linear or Monod-like equations are assumed and applied in the cellular model, N₂ fixation rates and intracellular Fe allocations which the model is parameterized and tested against are derived experimentally at only two Fe levels (Fig. 2). As the actual underlying relationships are unknown, it is possible that our model results may be off to a certain degree in particular at moderate levels of Fe. Nevertheless, the model reasonably fitted the measured growth rates observed at multiple Fe concentrations (i.e., 5 under ambient conditions and 4 under acidified conditions) (Fig. 3A), indicating that the assumed relationships in our model are reasonably sound.”

Figure 2C: should be anti-stress?

Corrected (Fig. 2C legend).

Figure 3: panel F (percent change) is not described in the figure legend

(D) and (E) in the figure caption should be (E) and (F), respectively. We have now corrected them.

Figure S6: should be ppmv, correct?

Corrected (Supplementary Fig. 6 legend).

Line 299: constraints, not constrains

Corrected (Line 328).

Cited References

Castruita, M., M. Saito, P. C. Schottel, L. A. Elmegreen, S. Myneni, E. I. Stiefel, and F. M. M. Morel (2006), Overexpression and characterization of an iron Storage and DNA-binding Dps protein from *Trichodesmium erythraeum*, *Appl. Environ. Microbiol.*, 72(4), 2918-2924, doi:10.1128/aem.72.4.2918-2924.2006.

Eichner, M., S. A. Kranz, and B. Rost (2014), Combined effects of different CO₂ levels and N sources on the diazotrophic cyanobacterium *Trichodesmium*, *Physiol. Plant.*, 152(2), 316-330, doi:10.1111/ppl.12172.

Harrison, P. M., and P. Arosio (1996), The ferritins: molecular properties, iron storage function and cellular regulation, *Biochimica et Biophysica Acta (BBA) - Bioenergetics*, 1275(3), 161-203, doi:10.1016/0005-2728(96)00022-9.

Hong, H., et al. (2017), The complex effects of ocean acidification on the prominent N₂-fixing cyanobacterium *Trichodesmium*, *Science*, 356(6337), 527-531, doi:10.1126/science.aal2981.

Hopkinson, B. M., C. L. Dupont, A. E. Allen, and F. M. M. Morel (2011), Efficiency of the CO₂-concentrating mechanism of diatoms, *Proceedings of the National Academy of Sciences*, 108(10), 3830-3837, doi:10.1073/pnas.1018062108.

Lal, K., et al. (2009), Proteomic comparison of four *Eimeria tenella* life-cycle stages: unsporulated oocyst, sporulated oocyst, sporozoite and second-generation merozoite, *Proteomics*, 9(19), 4566-4576, doi:10.1002/pmic.200900305.

Price, G. D., and M. R. Badger (1989), Expression of human carbonic anhydrase in the cyanobacterium *Synechococcus* PCC7942 creates a high CO₂-requiring phenotype, *Plant Physiol.*, 91(2), 505, doi:10.1104/pp.91.2.505.

Price, G. D., S.-i. Maeda, T. Omata, and M. R. Badger (2002), Modes of active inorganic carbon uptake in the cyanobacterium, *Synechococcus* sp. PCC7942 %J Functional Plant Biology, 29(3), 131-149, doi:10.1071/PP01229.

Shi, D., S. A. Kranz, J.-M. Kim, and F. M. M. Morel (2012), Ocean acidification slows nitrogen fixation and growth in the dominant diazotroph *Trichodesmium* under low-iron conditions,

Proceedings of the National Academy of Sciences, 109(45), E3094-E3100, doi:10.1073/pnas.1216012109.

Zeng, Y., X. P. Hu, G. Cao, and S. J. Suh (2018), Hemolymph protein profiles of subterranean termite *Reticulitermes flavipes* challenged with methicillin resistant *Staphylococcus aureus* or *Pseudomonas aeruginosa*, *Sci. Rep.*, 8(1), 13251, doi:10.1038/s41598-018-31681-2.

Zhang, S. F., Y. Zhang, Z. X. Xie, H. Zhang, L. Lin, and D. Z. Wang (2015), iTRAQ-based quantitative proteomic analysis of a toxigenic dinoflagellate *Alexandrium catenella* and its non-toxic mutant, *Proteomics*, 15(23-24), 4041-4050, doi:10.1002/pmic.201500156.

Reviewers' comments:

Reviewer #1 (Remarks to the Author):

The manuscript has been significantly improved by the authors' consideration of the reviewers' comments. I am grateful for the clarifications of the CCM and f(BC) questions that were raised, as well as the effort to address other concerns.

The most serious concern that I raised was that the model assumptions lead to conclusions that cells store significant quantities of iron while growing under iron limitation (for example, at 70% of maximum growth rate, a full 60% of intracellular iron is stored rather than used for propagating); I appreciate that storage was not the primary focus or motivation for this effort, but the particular finding raised my concern about whether this was a harbinger of shortcomings in the model that might also influence the main results and conclusions. Other reviewers raised valid points about model parameterization to a relatively sparse dataset which are now mentioned but cannot be materially addressed. I also find the authors' interpretation of concurrent high rates of storage with (ecologically relevant) seemingly self-imposed limitations in growth rate compelling enough that the conclusion should be less equivocal.

It seems I did not effectively communicate this concern, based on some of the responses. Included in the authors' response is, "We understand that the Reviewer was particularly concerned about the estimated high percentage of storage Fe under high Fe conditions.", but the concern was about the sub-optimal allocation towards storage (for future "rainy days") at the expense of cell growth in the present when iron is limiting. The general consistency between measurements of total cellular quota and select Fe-containing protein quotas collected under high Fe conditions in Shi et al and Hong et al. is comforting (but see below) but it seems more likely that differences under iron limitation between experiments could lead to an erroneous conclusion. The authors addressed this as much as possible by reporting similar growth rates under low iron conditions from Shi et al. and Hong et al.

Table 1 has the pCO₂ values backwards.

I believe that spectral count - based methods for quantification cannot achieve the detection limits implied here without employing additional metrics and data filtering.

Reviewer #2 (Remarks to the Author):

With this revised manuscript, the authors have thoroughly responded to my concerns and those of the other two reviewers. The manuscript has been substantially improved by the clarifications, additional information about sensitivity tests showing that their results were robust to certain specific assumptions, and the added discussion about the timescale of optimal intracellular resource allocation. I feel the revised manuscript better conveys important caveats and interesting future directions to continue this line of research.

I recommend publication of this revised manuscript.

Reviewer #3 (Remarks to the Author):

I reviewed the newest version of the manuscript, and read with interest the other reviews and replies. Regarding my original review I believe the authors have addressed my comments adequately. I think this is an exciting analysis, and while there are clearly limitations (laid out in the reviews) I believe the findings are robust. This version is better in that the limitations and assumptions are much more transparent to the reader. I'm sure there will still be differences of

opinion about whether the model is "correct" but I would like to see it published to stimulate those conversations.

Note our response to Reviewers' comments are in blue throughout this document.

Reviewer #1 (Remarks to the Author):

The manuscript has been significantly improved by the authors' consideration of the reviewers' comments. I am grateful for the clarifications of the CCM and f(BC) questions that were raised, as well as the effort to address other concerns.

We are glad that the reviewer found our manuscript substantially improved as we took the reviewers' comments into full consideration in revising the manuscript. We thank the reviewer for insightful and constructive comments that have been of great help to us in revising the manuscript.

The most serious concern that I raised was that the model assumptions lead to conclusions that cells store significant quantities of iron while growing under iron limitation (for example, at 70% of maximum growth rate, a full 60% of intracellular iron is stored rather than used for propagating); I appreciate that storage was not the primary focus or motivation for this effort, but the particular finding raised my concern about whether this was a harbinger of shortcomings in the model that might also influence the main results and conclusions. Other reviewers raised valid points about model parameterization to a relatively sparse dataset which are now mentioned but cannot be materially addressed. I also find the authors' interpretation of concurrent high rates of storage with (ecologically relevant) seemingly self-imposed limitations in growth rate compelling enough that the conclusion should be less equivocal. It seems I did not effectively communicate this concern, based on some of the responses. Included in the authors' response is, "We understand that the Reviewer was particularly concerned about the estimated high percentage of storage Fe under high Fe conditions.", but the concern was about the sub-optimal allocation towards storage (for future "rainy days") at the expense of cell growth in the present when iron is limiting. The general consistency between measurements of total cellular quota and select Fe-containing protein quotas collected under high Fe conditions in Shi et al and Hong et al. is comforting (but see below) but it seems more likely that differences under iron limitation between experiments could lead to an erroneous conclusion. The authors addressed this as much as possible by reporting similar growth rates under low iron conditions from Shi et al. and Hong et al.

We are very grateful that the reviewer appreciated our efforts in addressing his/her concerns on Fe storage in revising the manuscript. The sub-optimal allocation towards Fe

storage in *Trichodesmium* grown under intermediate Fe limitation (e.g., at 70% of maximum growth rate, 60% of intracellular Fe is stored, as the reviewer mentioned) is indeed intriguing and certainly worthy of further study. We note that previous laboratory studies on *Trichodesmium* also indicate that Fe storage very likely occurs despite the fact that growth is Fe limited. For instance, Kustka et al. [2003a, *Limnol. Oceanogr.*] and Berman-Frank et al. [2001, *Limnol. Oceanogr.*] have both observed that the growth rate of *Trichodesmium* increases linearly over a certain range of low intracellular Fe quota (i.e., severe Fe limitation), consistent with a lack of Fe storage; however, when Fe quota further increases while growth is still Fe limited (i.e., intermediate Fe limitation, growth rate < maximum growth rate), only a small portion of the increased intracellular Fe is put into use for growth and a significant fraction of it (~80%) appears to be stored. Moreover, using available physiological and biochemical data, Kustka et al. [2003b, *J. Phycol.*] estimate that even under Fe limitation as much as 44% of intracellular Fe in *Trichodesmium* can be stored.

These studies as well as our work demonstrate that considerable Fe storage in *Trichodesmium* growing under intermediate Fe limitation is possible. Although Fe storage is not the primary focus or motivation for our current study, as the reviewer appreciated, we are well aware of that such a phenomenon remains to be elucidated in future study, and have emphasized this in the revised manuscript (Lines 133-138, 141-142).

Table 1 has the pCO₂ values backwards.

Corrected. Thanks.

I believe that spectral count - based methods for quantification cannot achieve the detection limits implied here without employing additional metrics and data filtering.

We thank the reviewer for the comment, and would like to clarify that the statistical analysis has been appropriately performed for proteomic data. Figure R1 below illustrates the method we used to obtain the proteomic data, including the spectral counts in Supplementary Table S1 and in our previous “Response to reviewers’ comments (Table R4), and the differentially expressed proteins among experimental treatments in Hong et al. 2017.

Figure R1. Schematic illustration of an iTRAQ (multiplexed stable isotope tagging)-LC-MS/MS based method for proteomic analysis used in this manuscript and in Hong et al. 2017.

Basically, this is an iTRAQ (multiplexed stable isotope tagging)-LC-MS/MS based method, and in the “Bioinformatics analysis” step we have already conducted necessary statistical analysis and data filtering [i.e., peptide identification at the 95% confidence

interval is counted and filtered by 1% false discovery rate (FDR), and differential expression is analyzed by t-test and the results are further filtered using the Benjamini-Hochberg procedure for the FDR correction (5% FDR); for details, see Fig. R1 and Hong et al. 2017]. The obtained results include: 1] differentially expressed proteins among experimental treatments (e.g., ambient vs. acidified, high Fe vs. low Fe, etc.) with relative fold changes in protein abundance, and 2] spectral numbers (or called spectral counts in some literatures) of unique peptides/proteins (see Fig. R1). In the present manuscript, we used the spectral numbers of three abundant Fe-containing proteins, i.e., PsaC, ferritin and Dps, to roughly estimate (not quantify) their relative abundance (see columns 2 and 3 in Supplementary Table 1). In other words, the spectral number data and the data of differentially expressed proteins are from the same set of proteomic data for which statistical analysis and data filtering have been applied appropriately.

It has been shown that spectral counts (SC) have a good linear correlation with protein abundance, except in the case of low abundant proteins ($SC < 4$) [Lundgren et al. 2010]. However, all the three Fe-containing proteins investigated here (PsaC, ferritin, and Dps) are abundant proteins with $SC > 25$. Therefore, SC can be used to estimate the relative abundance of these three abundant proteins, although SC does have sensitivity issues with some low abundant proteins.

In some label-free shotgun proteomic approaches, spectral counting can be used to identify differentially expressed proteins between treatments, or to estimate **absolute** abundance of proteins. In those cases, suitable statistical models or modification or corrections of raw SC data are required [Vogel and Marcotte 2008; Lundgren et al. 2010]. However, in our study, we identified differentially expressed proteins by using iTRAQ labeling with necessary data filtering (see Fig. R1 and Hong et al. 2017), not the label-free method. In addition, the SC values of the three proteins we used in this manuscript (Supplementary Table 1) are simply for roughly estimating the **relative** abundance of the three proteins, as described above. Therefore, we believe no additional metrics and data filtering are required for this application.

In the previously submitted “Response to Reviewers’ comments”, we also showed SC values of a list of Fe-containing proteins (column 2 in Table R4). We did not use these SC data for further complex calculation or identification of the differentially expressed proteins. In this table, the data of relative fold changes (columns 3 to 5 in Table R4) are obtained by the above-mentioned iTRAQ labeling approach, which has already included necessary biostatistics analysis, not from the SC values. Therefore, we believe additional metrics or data filtering are not required either.

Please be noted that the absolute abundance of NifH, PsbA, PetC, and PsaC , which were used to estimate Fe allocation to nitrogenase and photosystems (Supplementary Table 3), were quantified by western blot (see Methods), not by proteomic analysis.

Again, we thank the reviewer for the comment, and have now clarified in the manuscript that the spectrum numbers were obtained with statistical analysis and data filtering (Supplementary Table 1).

Reviewer #2 (Remarks to the Author):

With this revised manuscript, the authors have thoroughly responded to my concerns and those of the other two reviewers. The manuscript has been substantially improved by the clarifications, additional information about sensitivity tests showing that their results were robust to certain specific assumptions, and the added discussion about the timescale of optimal intracellular resource allocation. I feel the revised manuscript better conveys important caveats and interesting future directions to continue this line of research.

I recommend publication of this revised manuscript.

Reviewer #3 (Remarks to the Author):

I reviewed the newest version of the manuscript, and read with interest the other reviews and replies. Regarding my original review I believe the authors have addressed my comments adequately. I think this is an exciting analysis, and while there are clearly limitations (laid out in the reviews) I believe the findings are robust. This version is better in that the limitations and assumptions are much more transparent to the reader. I'm sure there will still be differences of opinion about whether the model is "correct" but I would like to see it published to stimulate those conversations.

We are pleased that Reviewers #2 and #3 are now satisfied with our revised manuscript. We would like to take this opportunity to thank the reviewers for their very constructive comments that have greatly helped us in improving the manuscript.

References:

Berman-Frank, I., J. T. Cullen, Y. Shaked, R. M. Sherrell, and P. G. Falkowski (2001), Iron availability, cellular iron quotas, and nitrogen fixation in *Trichodesmium*, *Limnology and Oceanography*, 46(6), 1249-1260, doi:10.4319/lo.2001.46.6.1249.

Hong, H., R. Shen, F. Zhang, Z. Wen, S. Chang, W. Lin, S. A. Kranz, Y.-W. Luo, S.-J. Kao, F. M. M. Morel, and D. Shi (2017), The complex effects of ocean acidification on the prominent N₂-fixing cyanobacterium *Trichodesmium*, *Science*, 356(6337), 527-531, doi:10.1126/science.aal2981.

Kustka, A. B., S. A. Sanudo-Wilhelmy, E. J. Carpenter, D. G. Capone, J. Burns, and W. G. Sunda (2003a), Iron requirements for dinitrogen- and ammonium-supported growth in cultures of *Trichodesmium* (IMS 101): Comparison with nitrogen fixation rates and iron:carbon ratios of field populations, *Limnology and Oceanography*, 48(5), 1869-1884, doi:10.4319/lo.2003.48.5.1869.

Kustka, A., S. Sañudo-Wilhelmy, E. J. Carpenter, D. G. Capone, and J. A. Raven (2003b), A revised estimate of the iron use efficiency of nitrogen fixation, with special reference to the marine cyanobacterium *Trichodesmium* spp. (Cyanophyta), *Journal of Phycology*, 39(1), 12-25, doi:10.1046/j.1529-8817.2003.01156.x.

Lundgren DH, Hwang S, Wu L and Han DK (2010) Role of spectral counting in quantitative proteomics. *Expert. Rev. Proteomics* 7(1), 39-53

Vogel C and Marcotte EM (2008) Calculating absolute and relative protein abundance from mass spectrometry-based protein expression data. *Nature Protocols* 3, 1444-1451

REVIEWERS' COMMENTS:

Reviewer #1 (Remarks to the Author):

Regarding my comment on the growth rate limitation by iron and concomitant modeled iron storage, and your reply:

In your revised text on lines 133-138, I think it is appropriate to take credit for your interpretation of the data presented in citations 15 and 24; if I understand your interpretation correctly, it is distinct from that provided by the earlier authors and – if correct- could advance our understanding of iron storage in general.

At your discretion, consider rewording your text to include something like: "For example, two studies 15, 24 have both reported linear increases in growth rate over a range of low QFe values, consistent with a lack of Fe storage under severe limitation. However, marginal increases in QFe with further increases in Fe' (intermediate iron limitation) result in lesser increases in growth rate. We interpret these earlier data to suggest that only a small portion of the marginal increase in QFe is used for growth and that perhaps 80% of this marginal increase is used for storage; this interpretation is consistent with our model results.

Regarding my comment on spectral counting and your reply:

Thank you for the detailed and very helpful explanation of the methods used here. It is now clear to me that you used isobaric labeling for relative quantification of proteins among treatments and spectral counting for relative quantification among certain proteins. My earlier lack of understanding lead to my comment/concern about the reported detection limits and spectral counting based approaches.

Note our response to Reviewers' comments are in blue throughout this document.

REVIEWERS' COMMENTS:

Reviewer #1 (Remarks to the Author):

Regarding my comment on the growth rate limitation by iron and concomitant modeled iron storage, and your reply:

In your revised text on lines 133-138, I think it is appropriate to take credit for your interpretation of the data presented in citations 15 and 24; if I understand your interpretation correctly, it is distinct from that provided by the earlier authors and – if correct- could advance our understanding of iron storage in general.

We are pleased that Dr. Adam Kustka is now satisfied with the iron storage issue. We would also like to thank Adam again for his very thorough and constructive comments in the whole review process.

At your discretion, consider rewording your text to include something like: “For example, two studies 15, 24 have both reported linear increases in growth rate over a range of low Q_{Fe} values, consistent with a lack of Fe storage under severe limitation. However, marginal increases in Q_{Fe} with further increases in Fe' (intermediate iron limitation) result in lesser increases in growth rate. We interpret these earlier data to suggest that only a small portion of the marginal increase in Q_{Fe} is used for growth and that perhaps 80% of this marginal increase is used for storage; this interpretation is consistent with our model results.

The texts are now been revised following Adam's suggestion (Line 132-139):

For example, two studies ^{15,24} have both reported linear increases in growth rate of Trichodesmium over a range of low Q_{Fe} values, consistent with a lack of Fe storage under severe limitation. However, marginal increases in Q_{Fe} with further increases in inorganic Fe (Fe') (intermediate Fe limitation) result in lesser increases in growth rate. We interpret these earlier data to suggest that only a small portion of the marginal increase in Q_{Fe} is used for growth and that perhaps 80% of this marginal increase is used for storage; this interpretation is consistent with our model results.

Regarding my comment on spectral counting and your reply:

Thank you for the detailed and very helpful explanation of the methods used here. It is now clear to me that you used isobaric labeling for relative quantification of proteins among treatments and spectral counting for relative quantification among certain proteins. My earlier lack of understanding lead to my comment/concern about the reported detection limits and spectral counting based approaches.

We are glad that Adam accepted our explanations on spectral counting.